# Neural Sheaf Diffusion: A Topological Perspective on Heterophily and Oversmoothing in GNNs

**Cristian Bodnar**[*]
University of Cambridge
cristian.bodnar@cl.cam.ac.uk

**Francesco Di Giovanni**[†]
Twitter
fdigiovanni@twitter.com

**Benjamin P. Chamberlain**
Twitter

**Pietro Liò**
University of Cambridge

**Michael Bronstein**
University of Oxford & Twitter

## Abstract

Cellular sheaves equip graphs with a "geometrical" structure by assigning vector spaces and linear maps to nodes and edges. Graph Neural Networks (GNNs) implicitly assume a graph with a trivial underlying sheaf. This choice is reflected in the structure of the graph Laplacian operator, the properties of the associated diffusion equation, and the characteristics of the convolutional models that discretise this equation. In this paper, we use cellular sheaf theory to show that the underlying geometry of the graph is deeply linked with the performance of GNNs in heterophilic settings and their oversmoothing behaviour. By considering a hierarchy of increasingly general sheaves, we study how the ability of the sheaf diffusion process to achieve linear separation of the classes in the infinite time limit expands. At the same time, we prove that when the sheaf is non-trivial, discretised parametric diffusion processes have greater control than GNNs over their asymptotic behaviour. On the practical side, we study how sheaves can be learned from data. The resulting sheaf diffusion models have many desirable properties that address the limitations of classical graph diffusion equations (and corresponding GNN models) and obtain competitive results in heterophilic settings. Overall, our work provides new connections between GNNs and algebraic topology and would be of interest to both fields.

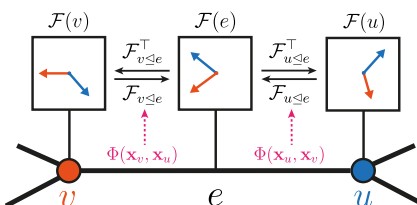

Figure 1: A sheaf $(G, \mathcal{F})$ shown for a single edge of the graph. The *stalks* are isomorphic to $\mathbb{R}^2$. The *restriction maps* $\mathcal{F}_{v \trianglelefteq e}$, $\mathcal{F}_{u \trianglelefteq e}$ and their adjoints move the vector features between these spaces. In practice, we learn the sheaf (i.e. the restrictions maps) from data via a parametric function $\Phi$.

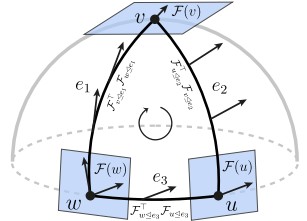

Figure 2: Analogy between parallel transport on a sphere and transport on a discrete vector bundle (cellular sheaf). A tangent vector is moved from $\mathcal{F}(w) \rightarrow \mathcal{F}(v) \rightarrow \mathcal{F}(u)$ and back. Because the vector returns in a different position, the transport is not path-independent.

---

[*]Work done as a research intern at Twitter.

[†]Proved the results in Section 3.1.

Our code is available at https://github.com/twitter-research/neural-sheaf-diffusion.

36th Conference on Neural Information Processing Systems (NeurIPS 2022).

# 1 Introduction

Graph Neural Networks (GNNs) [12, 20, 27–29, 39, 58, 64] have recently become very popular in the ML community as a model of choice to deal with relational and interaction data due to their multiple successful applications in domains ranging from social science and particle physics to structural biology and drug design. In this work, we focus on two main problems often observed in GNNs: their poor performance in heterophilic graphs [75] and their oversmoothing behaviour [48, 50]. The former arises from the fact that many GNNs are built on the strong assumption of *homophily*, i.e., that nodes tend to connect to other similar nodes. The latter refers to a phenomenon of some deeper GNNs producing features that are too smooth to be useful.

**Contributions.** We show that these two fundamental problems are linked by a common cause: the underlying "geometry" of the graph (used here in a very loose sense). When this geometry is trivial, as is typically the case, the two phenomena described above emerge. We make these statements precise through the lens of (cellular) sheaf theory [10, 18, 26, 44, 56, 62], a subfield of algebraic topology and geometry. Intuitively, a cellular sheaf associates a vector space to each node and edge of a graph, and a linear map between these spaces for each incident node-edge pair (Figure 1).

In Section 3, we analyse how by considering a hierarchy of increasingly general sheaves, starting from a trivial one, a diffusion equation based on the sheaf Laplacian [34] can solve increasingly more complicated node-classification tasks in the infinite time limit. In this regime, we show that oversmoothing and problems due to heterophily can be avoided by equipping the graph with the right sheaf structure for the task. In Section 4, we study the behaviour of a non-linear, parametric, and discrete version of this process. This results in a *Sheaf Convolutional Network* [32] that generalises Graph Convolutional Networks [39]. We prove that this discrete diffusion process is more flexible and has greater control over its asymptotic behaviour than GCNs [13, 51]. All these results are based on the properties of the harmonic space of the sheaf Laplacian, which we study from a spectral perspective in Section 3.1. We provide a new Cheeger-type inequality for the spectral gap of the sheaf Laplacian and note that these results might be of independent interest for spectral sheaf theory [34]. Finally, in Section 5, we apply our theory to designing simple and practical GNN models. We describe how to construct Sheaf Neural Networks by learning sheaves from data, thus making these types of models applicable beyond the toy experimental setting where they were originally introduced [32]. The resulting models obtain competitive results both in heterophilic and homophilic graphs.

# 2 Background

**Cellular Sheaves.** A *cellular sheaf* [18, 62] over a graph (Figure 1) is a mathematical object associating a vector space to each node and edge in the graph and a map between these spaces for each incident node-edge pair. We define this formally below:

**Definition 1.** *A cellular sheaf* $(G, \mathcal{F})$ *on an undirected graph* $G = (V, E)$ *consists of:*

- *A vector space* $\mathcal{F}(v)$ *for each* $v \in V$.
- *A vector space* $\mathcal{F}(e)$ *for each* $e \in E$.
- *A linear map* $\mathcal{F}_{v \trianglelefteq e} : \mathcal{F}(v) \to \mathcal{F}(e)$ *for each incident* $v \trianglelefteq e$ *node-edge pair.*

The vector spaces of the nodes and edges are called *stalks*, while the linear maps are referred to as *restriction maps*. The space formed by all the spaces associated with the nodes of the graph is called the space of 0-*cochains* $C^0(G; \mathcal{F}) := \bigoplus_{v \in V} \mathcal{F}(v)$, where $\bigoplus$ denotes the direct sum of vector spaces. For a 0-cochain $\mathbf{x} \in C^0(G; \mathcal{F})$, we use $\mathbf{x}_v$ to refer to the vector in $\mathcal{F}(v)$ of node $v$. Hansen and Ghrist [35] have constructed a convenient mental model for these objects based on opinion dynamics. In this context, $\mathbf{x}_v$ is the 'private opinion' of node $v$, while $\mathcal{F}_{v \trianglelefteq e} \mathbf{x}_v$ expresses how that opinion manifests publicly in a 'discourse space' formed by $\mathcal{F}(e)$. A particularly important subspace of $C^0(G; \mathcal{F})$ is the space of *global sections* $H^0(G; \mathcal{F}) := \{\mathbf{x} \in C^0(G; \mathcal{F}) : \mathcal{F}_{v \trianglelefteq e} \mathbf{x}_v = \mathcal{F}_{u \trianglelefteq e} \mathbf{x}_u\}$ containing those private opinions $\mathbf{x}$ for which all neighbours $(v, u)$ agree with each other in the discourse space. Given a cellular sheaf $(G, \mathcal{F})$, we can define a *sheaf Laplacian* operator [34] measuring the aggregated 'disagreement of opinions' at each node:

**Definition 2.** *The sheaf Laplacian of a sheaf* $(G, \mathcal{F})$ *is a linear map* $L_{\mathcal{F}} : C^0(G, \mathcal{F}) \to C^0(G, \mathcal{F})$ *defined node-wise as* $L_{\mathcal{F}}(\mathbf{x})_v := \sum_{v, u \trianglelefteq e} \mathcal{F}_{v \trianglelefteq e}^{\top}(\mathcal{F}_{v \trianglelefteq e} \mathbf{x}_v - \mathcal{F}_{u \trianglelefteq e} \mathbf{x}_u)$.

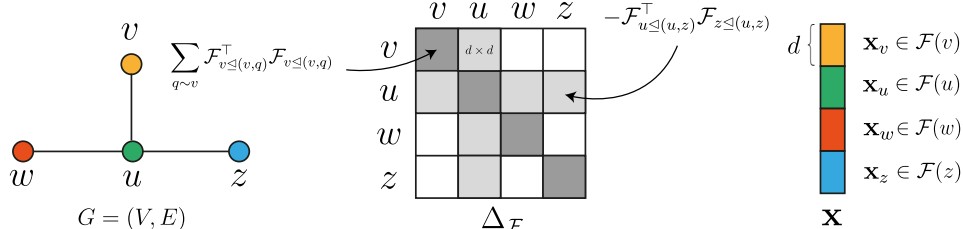

Figure 3: A graph *(left)*, the Laplacian matrix of a sheaf with $d$-dimensional stalks over the graph *(middle)* , and a 0-cochain $\mathbf{x}$ represented as a block-vector stacking the vectors of all nodes *(right)*.

The sheaf Laplacian is a positive semi-definite block matrix (Figure 3). The diagonal blocks are $L_{\mathcal{F}vv} = \sum_{v \trianglelefteq e} \mathcal{F}_{v \trianglelefteq e}^\top \mathcal{F}_{v \trianglelefteq e}$, while the non-diagonal blocks $L_{\mathcal{F}vu} = -\mathcal{F}_{v \trianglelefteq e}^\top \mathcal{F}_{u \trianglelefteq e}$. Denoting by $D$ the block-diagonal of $L_{\mathcal{F}}$, the normalised sheaf Laplacian is given by $\Delta_{\mathcal{F}} := D^{-1/2} L_{\mathcal{F}} D^{-1/2}$. For simplicity, we assume that all the stalks have a fixed dimension $d$. In that case, the sheaf Laplacian is a $nd \times nd$ real matrix, where $n$ is the number of nodes of $G$. When the vector spaces are set to $\mathbb{R}$ (i.e., $d = 1$ ) and the linear maps to the identity map over $\mathbb{R}$, the underlying sheaf is trivial and one recovers the well-known $n \times n$ graph Laplacian matrix and its normalised version $\Delta_0$. In general, $\Delta_{\mathcal{F}}$ is preferred to $L_{\mathcal{F}}$ for most practical purposes due to its bounded spectrum and, therefore, we focus on the former. A cochain $\mathbf{x}$ is called *harmonic* if $L_{\mathcal{F}}\mathbf{x} = 0$ or, equivalently, if $\mathbf{x} \in \ker(L_{\mathcal{F}})$. This means harmonic cochains are characterised by zero disagreements along all the edges of the graph, and it is not difficult to see that, in fact, $H^0(G; \mathcal{F})$ and $\ker(L_{\mathcal{F}})$ are isomorphic as vector spaces [35].

The sheaves with orthogonal maps (i.e. $\mathcal{F}_{v \trianglelefteq e} \in O(d)$ the Lie group of $d \times d$ orthogonal matrices) provide a more geometric interpretation of sheaves and play an important role in our analysis. Such sheaves are called *discrete $O(d)$ bundles* and can be seen as a discrete version of vector bundles [24, 60, 73] from differential geometry [67]. Intuitively, these objects describe vector spaces attached to the points of a manifold. In our discrete case, the role of the manifold is played by the graph, and the sheaf Laplacian describes how the elements of a vector space are transported via rotations in another neighbouring vector space similarly to how tangent vectors are moved across a manifold via parallel transport (connection; see Figure 2). Due to this analogy, the sheaf Laplacian on $O(d)$ *bundles* is also referred to as *connection Laplacian* [63].

**Heat Diffusion and GCNs.** Consider a graph with adjacency matrix $\mathbf{A}$, diagonal degree matrix $\mathbf{D}$, normalised graph Laplacian $\Delta_0 := \mathbf{I} - \mathbf{D}^{-1/2}\mathbf{A}\mathbf{D}^{-1/2}$, and an $n \times f$ feature matrix $\mathbf{X}$. We can define the heat diffusion equation and its Euler discretisation with a unit step as follows:

$$\dot{\mathbf{X}}(t) = -\Delta_0 \mathbf{X}(t) \ \rightsquigarrow \ \mathbf{X}(t+1) = \mathbf{X}(t) - \Delta_0 \mathbf{X}(t) = \textcolor{red}{(\mathbf{I} - \Delta_0)\mathbf{X}(t)}. \tag{1}$$

Comparing this with the Graph Convolutional Network [39] model, we observe that GCN is an augmented heat diffusion process with an additional $f \times f$ weight matrix $\mathbf{W}$ and a nonlinearity $\sigma$:

$$\mathrm{GCN}(\mathbf{X}, \mathbf{A}) := \sigma(\mathbf{D}^{-1/2}\mathbf{A}\mathbf{D}^{-1/2}\mathbf{X}\mathbf{W}) = \sigma(\textcolor{red}{(\mathbf{I} - \Delta_0)\mathbf{X}\mathbf{W}}). \tag{2}$$

From this perspective, it is perhaps not surprising that GCN is particularly affected by heterophily and oversmoothing since heat diffusion makes the features of neighbouring nodes increasingly smooth. In what follows, we consider a much more general and powerful family of (sheaf) diffusion processes leading to more expressive sheaf convolutions.

## 3 The Expressive Power of Sheaf Diffusion

**Preliminaries.** Let us now assume $G$ to be a graph with $d$-dimensional node feature vectors $\mathbf{x}_v \in \mathcal{F}(v)$. The features of all nodes are represented as a single vector $\mathbf{x} \in C^0(G; \mathcal{F})$ stacking all the individual $d$-dimensional vectors (Figure 3). Additionally, if we allow for $f$ feature channels, everything can be represented as a matrix $\mathbf{X} \in \mathbb{R}^{(nd) \times f}$, whose columns are vectors in $C^0(G; \mathcal{F})$. We are interested in the spatially discretised *sheaf diffusion* process governed by the following PDE:

$$\mathbf{X}(0) = \mathbf{X}, \quad \dot{\mathbf{X}}(t) = -\Delta_{\mathcal{F}}\mathbf{X}(t). \tag{3}$$

It can be shown that in the time limit, each feature channel is projected into $\ker(\Delta_{\mathcal{F}})$ [34]. As described above (up to a $D^{-1/2}$ normalisation), this space contains the signals that agree with the restriction maps of the sheaf along all the edges. Thus, sheaf diffusion can be seen as a 'synchronisation' process over the graph, where all the private opinions converge towards global agreement.

In this section, we investigate the expressive power of this process within the infinite time limit. Because the asymptotic behaviour of sheaf diffusion is determined by the properties of $\ker(\Delta_{\mathcal{F}})$, in Section 3.1, we investigate when this subspace is non-trivial (i.e. it contains more than just the zero vector). In Section 3.2, we use this characterisation of the harmonic space to study what sort of sheaf diffusion processes will asymptotically produce projections into $\ker(\Delta_{\mathcal{F}})$ that can linearly separate the classes for various kinds of graphs and initial conditions. Since diffusion converges exponentially fast, the following results are also relevant for models with finite integration time or layers.

### 3.1 Harmonic Space of Sheaf Laplacians

A major role in the analysis below is played by discrete vector bundles, and we concentrate on this case. We note though that our results below generalise to the general linear group $\mathcal{F}_{v \trianglelefteq e} \in GL(d)$, the Lie group of $d \times d$ invertible matrices, provided we can also control the norm of the restriction maps from below. Given a discrete $O(d)$-bundle, $\mathcal{F}_{v \trianglelefteq e}^{\top} \mathcal{F}_{v \trianglelefteq e} = \mathbf{I}_d$ and the block diagonal of $L_{\mathcal{F}}$ has a diagonal structure since $L_{\mathcal{F}vv} = d_v \mathbf{I}_d$, where $d_v$ is the degree of node $v$. Accordingly, if a signal $\tilde{\mathbf{x}} \in \ker(L_{\mathcal{F}})$, then the signal $\mathbf{x} : v \mapsto \sqrt{d_v} \tilde{\mathbf{x}}_v \in \ker(\Delta_{\mathcal{F}})$ and similarly for the inverse transformation.

Key to our analysis is studying *transport* operators induced by the restriction maps of the sheaf. Given nodes $v, u \in V$ and a path $\gamma_{v \to u} = (v, v_1, \ldots, v_\ell, u)$ from $v$ to $u$, we consider a notion of *transport* from the stalk $\mathcal{F}(v)$ to the stalk $\mathcal{F}(u)$, constructed by composing restriction maps (and their transposes) along the edges:

$$\mathbf{P}_{v \to u}^{\gamma} := (\mathcal{F}_{u \trianglelefteq e}^{\top} \mathcal{F}_{v_\ell \trianglelefteq e}) \ldots (\mathcal{F}_{v_1 \trianglelefteq e}^{\top} \mathcal{F}_{v \trianglelefteq e}) : \mathcal{F}(v) \to \mathcal{F}(u).$$

For general sheaf structures, the graph transport is *path dependent*, meaning that how the vectors are transported across two nodes depends on the path between them (see Figure 2). In fact, we show that this property characterises the *spectral gap* of a sheaf Laplacian, i.e. the smallest eigenvalue of $\Delta_{\mathcal{F}}$.

**Proposition 3.** *If $\mathcal{F}$ is a discrete $O(d)$ bundle over a connected graph and $r :=$ $\max_{\gamma_{v \to u}, \gamma_{v \to u}'} ||\mathbf{P}_{v \to u}^{\gamma} - \mathbf{P}_{v \to u}^{\gamma'}||$, then we have $\lambda_0^{\mathcal{F}} \leq r^2/2$.*

A consequence of this result is that there is always a non-trivial harmonic space (i.e. $\lambda_0^{\mathcal{F}} = 0$) if the transport maps generated by an orthogonal sheaf are *path-independent* (i.e. $r = 0$). Next, we address the opposite direction.

**Proposition 4.** *If $\mathcal{F}$ is a discrete $O(d)$ bundle over a connected graph and $\mathbf{x} \in H^0(G, \mathcal{F})$, then for any cycle $\gamma$ based at $v \in V$ we have $\mathbf{x}_v \in \ker(\mathbf{P}_{v \to v}^{\gamma} - \mathbf{I})$.*

This proposition highlights the interplay between the graph and the sheaf structure. A simple consequence of this result is that for any cycle-free subset $S \subset V$, we have that any sheaf (or connection-) Laplacian restricted to $S$ always admits a non-trivial harmonic space. A natural question connected to the previous result is whether a Cheeger-like inequality holds in the other direction. This turns out to be the case:

**Proposition 5.** *Let $\mathcal{F}$ be a discrete $O(d)$ bundle over a connected graph $G$ with $n$ nodes and let $||(\mathbf{P}_{v \to v}^{\gamma} - \mathbf{I})\mathbf{x}_v|| \geq \epsilon ||\mathbf{x}_v||$ for all cycles $\gamma_{v \to v}$. Then $\lambda_0^{\mathcal{F}} \geq \epsilon^2 (2\mathrm{diam}(G) n \, d_{max})^{-1}$.*

While the bound above is of little use in practice, it shows how the spectral gap of a sheaf Laplacian is indeed related to the deviation of the transport maps from being path-independent, as measured by $\epsilon$. We note that the Cheeger-like inequality presented here is not unique, and other types of bounds on $\lambda_0^{\mathcal{F}}$ have been derived [2]. We conclude this section by further analysing the dimensionality of the harmonic space of discrete $O(d)$-bundles:

**Lemma 6.** *Let $\mathcal{F}$ be a discrete $O(d)$ bundle over a connected graph $G$. Then $\dim(H^0) \leq d$ and $\dim(H^0) = d$ if and only if the transport is path-independent.*

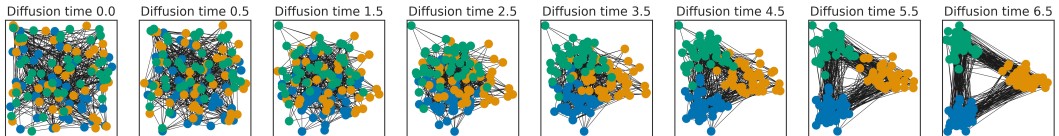

Figure 4: Diffusion process on $O(2)$-bundles progressively separates the classes of the graph.

## 3.2 The Linear Separation Power of Sheaf Diffusion

In what follows, we use the results above to analyse the ability of certain classes of sheaves to linearly separate the features in the limit of the diffusion processes they induce. We utilise this as a proxy for the capacity of certain diffusion processes to avoid oversmoothing.

**Definition 7.** *A hypothesis class of sheaves with $d$-dimensional stalks $\mathcal{H}^d$ has* linear separation power *over a family of graphs $\mathcal{G}$ if for any labelled graph $G = (V, E) \in \mathcal{G}$, there is a sheaf $(\mathcal{F}, G) \in \mathcal{H}^d$ that can linearly separate the classes of $G$ in the time limit of Equation 3 for almost all initial conditions.*

Note that the restriction to almost all initial conditions is necessary because, in the limit, diffusion behaves like a projection in the harmonic space and there will always be degenerate initial conditions (e.g. the zero matrix) that will yield a zero projection. We will now show how the choice of the sheaf impacts the behaviour of the diffusion process. For this purpose, we will consider a hierarchy of increasingly general classes of sheaves.

**Symmetric invertible.** $\mathcal{H}^d_{\mathrm{sym}} := \{(\mathcal{F}, G) : \mathcal{F}_{v \trianglelefteq e} = \mathcal{F}_{u \trianglelefteq e}, \ \det(\mathcal{F}_{v \trianglelefteq e}) \neq 0\}$. We note that for $d = 1$, the sheaf Laplacians induced by this class of sheaves coincides with the set of the well-known weighted graph Laplacians with strictly positive weights, which also includes the usual graph Laplacian (see proof in Appendix B). Therefore, this hypothesis class is of particular interest since it includes those graph Laplacians typically used by graph convolutional models such as GCN [39] and ChebNet [20]. We first show that this class of sheaf Laplacians can linearly separate the classes in binary classification settings under certain homophily assumptions:

**Proposition 8.** *Let $\mathcal{G}$ be the set of connected graphs $G = (V, E)$ with two classes $A, B \subset V$ such that for each $v \in A$, there exists $u \in A$ and an edge $(v, u) \in E$. Then $\mathcal{H}^1_{\mathrm{sym}}$ has linear separation power over $\mathcal{G}$.*

In contrast, under certain heterophilic conditions, this hypothesis class is not powerful enough to linearly separate the two classes no matter what the initial conditions are:

**Proposition 9.** *Let $\mathcal{G}$ be the set of connected bipartite graphs $G = (A, B, E)$, with partitions $A, B$ forming two classes and $|A| = |B|$. Then $\mathcal{H}^1_{\mathrm{sym}}$ cannot linearly separate the classes of any graph in $\mathcal{G}$ for any initial conditions $\mathbf{X}(0) \in \mathbb{R}^{n \times f}$.*

**Non-symmetric invertible.** $\mathcal{H}^d := \{(\mathcal{F}, G) : \det(\mathcal{F}_{v \trianglelefteq e}) \neq 0\}$. This larger hypothesis class addresses the above limitation by allowing non-symmetric relations:

**Proposition 10.** *Let $\mathcal{G}$ contain all the connected graphs $G = (V, E)$ with two classes $A, B \subseteq V$. Consider a sheaf $(\mathcal{F}; G) \in \mathcal{H}^1$ with $\mathcal{F}_{v \trianglelefteq e} = -\alpha_e$ if $v \in A$ and $\mathcal{F}_{u \trianglelefteq e} = \alpha_e$ if $u \in B$ with $\alpha_e > 0$ for all $e \in E$. Then the diffusion induced by $(\mathcal{F}; G)$ can linearly separate the classes of $G$ for almost all initial conditions, and $\mathcal{H}^1$ has linear separation power over $\mathcal{G}$.*

Since $\mathcal{F}_{v \trianglelefteq e}^\top \mathcal{F}_{u \trianglelefteq e} = \pm \alpha_e^2$, the type of sheaf above can be interpreted as a discrete $O(1)$-bundle over a weighted graph with edge weights $\alpha_e^2$ and transport maps $\mathcal{F}_{v \trianglelefteq e}^\top \mathcal{F}_{u \trianglelefteq e} = -1$ for the inter-class edges and $+1$ for the intra-class edges. Intuitively, this type of transport, which is path-independent, polarises the features of the two classes and forces them to take opposite signs in the infinite limit. This provides a sheaf-theoretic explanation for why negatively-weighted edges have been widely adopted in heterophilic settings [7, 17, 72].

So far we have only studied the effects of changing the type of sheaves in dimension one. We now consider the effects of adjusting the dimension of the stalks and begin by stating a fundamental limitation of (sheaf) diffusion when $d = 1$.

**Proposition 11.** *Let $G$ be a connected graph with $C \geq 3$ classes. Then, $\mathcal{H}^1$ cannot linearly separate the classes of $G$ for any initial conditions $\mathbf{X}(0) \in \mathbb{R}^{n \times f}$.*

This is essentially a consequence of $\dim\big(\ker(\Delta_{\mathcal{F}})\big) \leq 1$ in this case, by virtue of Lemma 6. From a GNN perspective, this means that in the infinite depth setting, sufficient *stalk width* (i.e., dimension $d$) is needed in order to solve tasks involving more than two classes. Note that $d$ is different from the classical notion of feature channels $f$. As the result above shows, the latter has no effect on the linear separability of the classes in $d = 1$. Next, we will see that the former does.

**Diagonal invertible.** $\mathcal{H}^d := \{(\mathcal{F}, G) : \text{diagonal } \mathcal{F}_{v\trianglelefteq e}, \det(\mathcal{F}_{v\trianglelefteq e}) \neq 0\}$. The sheaves in this class can be seen as $d$ independent sheaves from $\mathcal{H}^1$ encoded in the $d$-dimensional diagonals of their restriction maps. This perspective allows us to generalise Proposition 10 to a multi-class setting:

**Proposition 12.** *Let $\mathcal{G}$ be the set of connected graphs with nodes belonging to $C \geq 3$ classes. Then for $d \geq C$, $\mathcal{H}^d_{\mathrm{diag}}$ has linear separation power over $\mathcal{G}$.*

This result illustrates the benefits of using higher-dimensional stalks while maintaining a simple and computationally convenient class of diagonal restriction maps. Next, with more complex restriction maps, we can show that lower-dimensional stalks can be used to achieve linear separation in the presence of even more classes.

**Orthogonal.** $\mathcal{H}^d_{\mathrm{orth}} := \{(\mathcal{F}, G) : \mathcal{F}_{v\trianglelefteq e} \in O(d)\}$ is the class of $O(d)$-bundles. Orthogonal maps are able to make more efficient use of the space available to them than diagonal restriction maps:

**Proposition 13.** *Let $\mathcal{G}$ be the class of connected graphs with $C \leq 2d$ classes. Then, for all $d \in \{2, 4\}$, $\mathcal{H}^d_{\mathrm{orth}}$ has linear separation power over $\mathcal{G}$.*

Figure 4 includes an example diffusion process over an $O(2)$-bundle.

**Summary:** Different sheaf classes give rise to different behaviours of the diffusion process and, consequently, to different separation capabilities. Taken together, these results show that solving any node classification task can be reduced to performing diffusion with the right sheaf.

## 4 Expressive Power of Sheaf Convolutions

Analogously to how GCN augments heat diffusion, we can construct a **Sheaf Convolutional Network (SCN)** augmenting the sheaf diffusion process. In this section, we analyse the capacity of SCNs to change, *if necessary*, their asymptotic behaviour compared to the base diffusion process. Since the sheaf structure will be ultimately learned from data, this is particularly important for the common setting when the learned sheaf is different from the "ground truth" sheaf for the task to be solved.

The continous diffusion process from Equation 3 has the Euler discretisation with unit step-size $\mathbf{X}(t + 1) = \mathbf{X}(t) - \Delta_{\mathcal{F}}\mathbf{X}(t) = (\mathbf{I}_{nd} - \Delta_{\mathcal{F}})\mathbf{X}(t)$. Assuming $\mathbf{X} \in \mathbb{R}^{nd \times f_1}$, we can equip the right side with weight matrices $\mathbf{W}_1 \in \mathbb{R}^{d \times d}$, $\mathbf{W}_2 \in \mathbb{R}^{f_1 \times f_2}$ and a non-linearity $\sigma$ to arrive at the following model originally proposed by Hansen and Gebhart [32]:

$$\mathbf{Y} = \sigma\Big(\big(\mathbf{I}_{nd} - \Delta_{\mathcal{F}}\big)(\mathbf{I}_n \otimes \mathbf{W}_1)\mathbf{X}\mathbf{W}_2\Big) \in \mathbb{R}^{nd \times f_2}, \tag{4}$$

where $f_1, f_2$ are the number of input and output feature channels, and $\otimes$ denotes the Kronecker product. Here, $\mathbf{W}_1$ multiplies from the left the vector feature of all the nodes in all channels (i.e. $\mathbf{W}_1\mathbf{x}_v^i$ for all $v$ and channels $i$), while $\mathbf{W}_2$ multiplies the features from the right and can adjust the number of feature channels, just like in GCNs. As one would expect, when using a trivial sheaf, $\Delta_{\mathcal{F}} = \Delta_0$, $\mathbf{W}_1$ becomes a scalar and one recovers the GCN of Kipf and Welling [39]. To see how SCNs behave compared to their base diffusion process, we investigate how SCN layers affect the *sheaf Dirichlet energy* $E_{\mathcal{F}}(\mathbf{x})$, which sheaf diffusion is known to minimise over time.

**Definition 14.** $E_{\mathcal{F}}(\mathbf{x}) := \mathbf{x}^\top \Delta_{\mathcal{F}}\mathbf{x} = \frac{1}{2}\sum_{e:=(v,u)}\|\mathcal{F}_{v\trianglelefteq e}D_v^{-1/2}\mathbf{x}_v - \mathcal{F}_{u\trianglelefteq e}D_u^{-1/2}\mathbf{x}_u\|_2^2$

Similarly, for multiple channels the energy is $E_{\mathcal{F}}(\mathbf{X}) := \mathrm{trace}(\mathbf{X}^\top \Delta_{\mathcal{F}}\mathbf{X})$. This is a measure of how close a signal $\mathbf{x}$ is to $\ker(\Delta_{\mathcal{F}})$ and it is easy to see that $\mathbf{x} \in \ker(\Delta_{\mathcal{F}}) \Leftrightarrow E_{\mathcal{F}}(\mathbf{x}) = 0$. We begin by studying the sheaves for which the energy decreases and representations end up asymptotically in $\ker(\Delta_{\mathcal{F}})$. Let $\lambda_* := \max_{i>0}(\lambda_i^{\mathcal{F}} - 1)^2 \leq 1$ and denote by $\mathcal{H}^1_+ := \{(\mathcal{F}, G) \mid \mathcal{F}_{v\trianglelefteq e}\mathcal{F}_{u\trianglelefteq e} > 0\}$.

**Theorem 15.** *For $(\mathcal{F}, G) \in \mathcal{H}^1_+$ and $\sigma$ being (Leaky)ReLU, $E_{\mathcal{F}}(\mathbf{Y}) \leq \lambda_*\|\mathbf{W}_1\|_2^2\|\mathbf{W}_2^\top\|_2^2 E_{\mathcal{F}}(\mathbf{X})$.*

This generalises existent results for GCNs [13, 51] and proves that SCNs using this family of Laplacians, which includes all weighted graph Laplacians, exponentially converge to $\ker(\Delta_{\mathcal{F}})$ if $\lambda_* \|\mathbf{W}_1\|_2^2 \|\mathbf{W}_2^\top\|_2^2 < 1$. In particular, if $E_{\mathcal{F}}(\mathbf{X}) = 0$, then $E_{\mathcal{F}}(\mathbf{Y}) = 0$ and the representations remain trapped inside the kernel no matter what the norm of the weights is. Therefore, in settings as those described by Propositions 9 and 11, the linear separation capabilities of this class of models are severely limited (see Corollaries 36, 37 in Appendix B).

Finally, the Theorem also extends to bundles with symmetric maps, $\mathcal{H}_{\mathrm{orth,sym}}^d := \mathcal{H}_{\mathrm{orth}}^d \cap \mathcal{H}_{\mathrm{sym}}^d$:

**Theorem 16.** *If $(\mathcal{F}, G) \in \mathcal{H}_{\mathrm{orth,sym}}^d$ and $\sigma = (Leaky)ReLU$, $E_{\mathcal{F}}(\mathbf{Y}) \leq \lambda_* \|\mathbf{W}_1\|_2^2 \|\mathbf{W}_2^\top\|_2^2 E_{\mathcal{F}}(\mathbf{X})$.*

In some sense, this is not surprising because, for this class, $\ker(\Delta_{\mathcal{F}})$ contains the same information as the kernel of the classical normalised graph Laplacian (see Proposition 29 in Appendix C).

More generally, SCNs with sheaves outside $\mathcal{H}_{\mathrm{sym}}^d$, are much more flexible and can easily increase the Dirichlet energy using an arbitrarily small linear transformation $\mathbf{W}_1$:

**Proposition 17.** *For any connected graph $G$ and $\varepsilon > 0$, there exist a sheaf $(G, \mathcal{F}) \notin \mathcal{H}_{\mathrm{sym}}^d$, $\mathbf{W}_1$ with $\|\mathbf{W}_1\|_2 < \varepsilon$ and feature vector $\mathbf{x}$ such that $E_{\mathcal{F}}((\mathbf{I} \otimes \mathbf{W}_1)\mathbf{x}) > E_{\mathcal{F}}(\mathbf{x})$.*

Importantly, this proves that this family of SCNs can, if necessary, escape the kernel of the Laplacian.

> **Summary:** Not only that sheaf diffusion is more expressive than heat diffusion as shown in Section 3.2, but SCNs are also more expressive than GCNs in the sense that they are generally not constrained to decrease the Dirichlet energy when using low-norm weights. This provides them with greater control than GCNs over their asymptotic behaviour.

## 5 Neural Sheaf Diffusion and Sheaf Learning

In the previous sections, we discussed the various advantages provided by sheaf diffusion and sheaf convolutions. However, in general, the ground truth sheaf is unknown or unspecified. Therefore, we aim to learn the underlying sheaf from data end-to-end, thus allowing the model to pick the right geometry for solving the task.

**Neural Sheaf Diffusion.** We propose the diffusion-type model from Equation 5. We note that by setting $\mathbf{W}_1, \mathbf{W}_2$ to identity and $\sigma(\mathbf{x}) = \mathrm{ELU}(\epsilon\mathbf{x})/\epsilon$ with $\epsilon > 0$ small enough or simply $\sigma = \mathrm{id}$, we recover (up to a scaling) the sheaf diffusion equation. Therefore, the model is at least as expressive as sheaf diffusion and benefits from all the positive properties outlined in Section 3.2.

$$\dot{\mathbf{X}}(t) = -\sigma\Big(\Delta_{\mathcal{F}(t)}(\mathbf{I}_n \otimes \mathbf{W}_1)\mathbf{X}(t)\mathbf{W}_2\Big), \tag{5}$$

Crucially, the sheaf Laplacian $\Delta_{\mathcal{F}(t)}$ is that of a sheaf $(G, \mathcal{F}(t))$ that *evolves over time*. More specifically, the evolution of the sheaf structure is described by a learnable function of the data $(G, \mathcal{F}(t)) = g(G, \mathbf{X}(t); \theta)$. This allows the model to use the latest available features to manipulate the underlying geometry of the graph and, implicitly, the behaviour of the diffusion process. Additionally, We use an MLP followed by a reshaping to map the raw features of the dataset to a matrix $\mathbf{X}(0)$ of shape $nd \times f$ and a final linear layer to perform the node classification.

In our experiments, we focus on the time-discretised version of this model from Equation 6, which allows us to use a new set of weights at each layer $t$ while maintaining the nice theoretical properties of the model above.

$$\mathbf{X}_{t+1} = \mathbf{X}_t - \sigma\Big(\Delta_{\mathcal{F}(t)}(\mathbf{I} \otimes \mathbf{W}_1^t)\mathbf{X}_t\mathbf{W}_2^t\Big) \tag{6}$$

We note that this model is different from the SCN model from Equation 4 in two major ways. First, Hansen and Gebhart [32] used a *hand-crafted* sheaf with $d = 1$, constructed in a synthetic setting with full knowledge of the data-generating process. In contrast, we *learn* a sheaf, which makes our model applicable to any real-world graph dataset, even in the absence of a sheaf structure. Additionally, motivated by our theoretical results, we use the full generality of sheaves by using stalks with $d \geq 1$ and higher-dimensional maps. Second, our model uses a residual parametrisation of the discretised diffusion process, which empirically improves its performance.

**Sheaf Learning.** The restriction maps are learned using *locally* available information. Each $d \times d$ matrix $\mathcal{F}_{v \trianglelefteq e}$ is learned via a parametric matrix-valued function $\Phi$, with $\mathcal{F}_{v \trianglelefteq e := (v,u)} = \Phi(\mathbf{x}_v, \mathbf{x}_u)$.

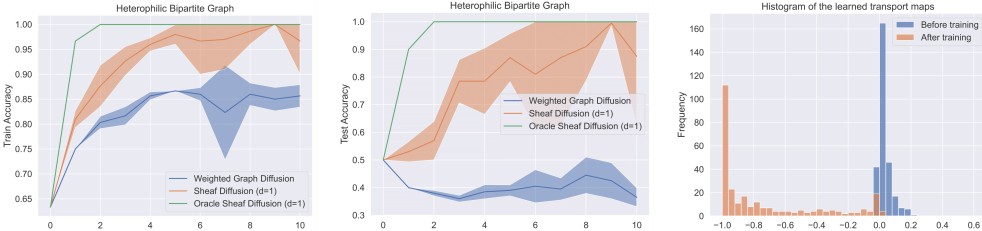

Figure 5: (*Left*) Train and (*Middle*) test accuracy as a function of diffusion time. (*Right*) Histogram of the learned scalar transport maps. The performance of the sheaf diffusion model is superior to that of weighted-graph diffusion and correctly learns to invert the features of the two classes.

This function must be non-symmetric to be able to learn asymmetric transport maps along each edge. In practice, we set $\Phi(\mathbf{x}_v, \mathbf{x}_u) = \sigma(\mathbf{V}[\mathbf{x}_v||\mathbf{x}_u])$ followed by a reshaping of the output, where $\mathbf{V}$ is a weight matrix. For simplicity, the equations above use a single feature channel, but in practice, all channels are supplied as input. More generally, we can show that if the function $\Phi$ has enough capacity and the features are diverse enough, we can learn any sheaf over a graph.

**Proposition 18.** *Let $G = (V, E)$ be a finite graph with features $\mathbf{X}$. Then, if $(\mathbf{x}_v, \mathbf{x}_u) \neq (\mathbf{x}_w, \mathbf{x}_z)$ for any $(v, u) \neq (w, z) \in E$ and $\Phi$ is an MLP with sufficient capacity, $\Phi$ can learn any sheaf $(\mathcal{F}; G)$.*

First, this result formally motivates learning a sheaf at each layer since the model can learn to distinguish more nodes after each aggregation step. Second, this suggests that more expressive models (in the Weisfeiler-Lehman sense [8, 9, 46, 71]) could learn a more general family of sheaves. We leave a deeper investigation of these aspects for future work. In what follows, we distinguish between several types of functions $\Phi$ depending on the type of matrix they learn.

**Diagonal.** The main advantage of this parametrisation is that fewer parameters need to be learned per edge, and the sheaf Laplacian ends up being a matrix with diagonal blocks, which also results in fewer operations in sparse matrix multiplications. The main disadvantage is that the $d$ dimensions of the stalks interact only via the left $\mathbf{W}_1$ multiplication.

**Orthogonal.** In this case, the model effectively learns a discrete vector bundle. Orthogonal matrices provide several advantages: (1) they can mix the various dimension of the stalks, (2) the orthogonality constraint prevents overfitting while reducing the number of parameters, (3) they have better understood theoretical properties, and (4) the resulting Laplacians are easier to normalise numerically since the diagonal entries correspond to the degrees of the nodes. In our model, we build orthogonal matrices from a composition of Householder reflections [45].

**General.** Finally, we consider the most general option of learning arbitrary matrices. The maximal flexibility these maps provide can be useful, but it also comes with the danger of overfitting. At the same time, the sheaf Laplacian is more challenging to normalise numerically since one has to compute $D^{-1/2}$ for a positive semi-definite matrix $D$. To perform this at scale, one has to rely on SVD, whose gradients can be infinite if $D$ has repeated eigenvalues. Therefore, this model is more challenging to train.

**Computational Complexity.** The GCN from Equation 2 has complexity $\mathcal{O}(nc^2 + mc)$, where $c$ is the number of channels and $m$ the number of edges. Assume a sheaf diffusion model with stalk dimension $d$ and $f$ channels such that $d \times f = c$ (i.e. same representation size). Then, when the model uses diagonal maps, the complexity is $\mathcal{O}(nc^2 + mdc)$. When using orthogonal or general matrices, the complexity becomes $\mathcal{O}(n(c^2 + d^3) + m(cd^2 + d^3))$ (see Appendix E.1 for detailed derivations). In practice, we use $1 \leq d \leq 5$, which effectively results in a constant overhead compared to GCN.

## 6 Experiments

**Synthetic experiments.** We consider a simple setup given by a connected bipartite graph with equally sized partitions. We sample the features from two overlapping isotropic Gaussian distributions to make the classes linearly non-separable at initialisation time. From Proposition 9, we know that diffusion models using symmetric restriction maps cannot separate the classes in the limit, while a diffusion process using negative transport maps can. Therefore, we use two vanilla sheaf diffusion processes by setting $d = 1$, $\mathbf{W}_1 = \mathbf{I}_d$, $\mathbf{W}_2 = \mathbf{I}_f$ and $\sigma = \mathrm{id}$ in Equation 5. In both models, we learn

Table 1: Results on node classification datasets sorted by their homophily level. Top three models are coloured by **First**, **Second**, **Third**. Our models are marked **NSD**.

| | Texas | Wisconsin | Film | Squirrel | Chameleon | Cornell | Citeseer | Pubmed | Cora |
|---|---|---|---|---|---|---|---|---|---|
| Hom level | 0.11 | 0.21 | 0.22 | 0.22 | 0.23 | 0.30 | 0.74 | 0.80 | 0.81 |
| #Nodes | 183 | 251 | 7,600 | 5,201 | 2,277 | 183 | 3,327 | 18,717 | 2,708 |
| #Edges | 295 | 466 | 26,752 | 198,493 | 31,421 | 280 | 4,676 | 44,327 | 5,278 |
| #Classes | 5 | 5 | 5 | 5 | 5 | 5 | 7 | 3 | 6 |
| **Diag-NSD** | $85.67_{\pm6.95}$ | $88.63_{\pm2.75}$ | $37.79_{\pm1.01}$ | $54.78_{\pm1.81}$ | $68.68_{\pm1.73}$ | $86.49_{\pm7.35}$ | $77.14_{\pm1.85}$ | $89.42_{\pm0.43}$ | $87.14_{\pm1.06}$ |
| **O(d)-NSD** | $85.95_{\pm5.51}$ | $89.41_{\pm4.74}$ | $37.81_{\pm1.15}$ | $56.34_{\pm1.32}$ | $68.04_{\pm1.58}$ | $84.86_{\pm4.71}$ | $76.70_{\pm1.57}$ | $89.49_{\pm0.40}$ | $86.90_{\pm1.13}$ |
| **Gen-NSD** | $82.97_{\pm5.13}$ | $89.21_{\pm3.84}$ | $37.80_{\pm1.22}$ | $53.17_{\pm1.31}$ | $67.93_{\pm1.58}$ | $85.68_{\pm6.51}$ | $76.32_{\pm1.65}$ | $89.33_{\pm0.35}$ | $87.30_{\pm1.15}$ |
| GGCN | $84.86_{\pm4.55}$ | $86.86_{\pm3.29}$ | $37.54_{\pm1.56}$ | $55.17_{\pm1.58}$ | $71.14_{\pm1.84}$ | $85.68_{\pm6.63}$ | $77.14_{\pm1.45}$ | $89.15_{\pm0.37}$ | $87.95_{\pm1.05}$ |
| H2GCN | $84.86_{\pm7.23}$ | $87.65_{\pm4.98}$ | $35.70_{\pm1.00}$ | $36.48_{\pm1.86}$ | $60.11_{\pm2.15}$ | $82.70_{\pm5.28}$ | $77.11_{\pm1.57}$ | $89.49_{\pm0.38}$ | $87.87_{\pm1.20}$ |
| GPRGNN | $78.38_{\pm4.36}$ | $82.94_{\pm4.21}$ | $34.63_{\pm1.22}$ | $31.61_{\pm1.24}$ | $46.58_{\pm1.71}$ | $80.27_{\pm8.11}$ | $77.13_{\pm1.67}$ | $87.54_{\pm0.38}$ | $87.95_{\pm1.18}$ |
| FAGCN | $82.43_{\pm6.89}$ | $82.94_{\pm7.95}$ | $34.87_{\pm1.25}$ | $42.59_{\pm0.79}$ | $55.22_{\pm3.19}$ | $79.19_{\pm9.79}$ | N/A | N/A | N/A |
| MixHop | $77.84_{\pm7.73}$ | $75.88_{\pm4.90}$ | $32.22_{\pm2.34}$ | $43.80_{\pm1.48}$ | $60.50_{\pm2.53}$ | $73.51_{\pm6.34}$ | $76.26_{\pm1.33}$ | $85.31_{\pm0.61}$ | $87.61_{\pm0.85}$ |
| GCNII | $77.57_{\pm3.83}$ | $80.39_{\pm3.40}$ | $37.44_{\pm1.30}$ | $38.47_{\pm1.58}$ | $63.86_{\pm3.04}$ | $77.86_{\pm3.79}$ | $77.33_{\pm1.48}$ | $90.15_{\pm0.43}$ | $88.37_{\pm1.25}$ |
| Geom-GCN | $66.76_{\pm2.72}$ | $64.51_{\pm3.66}$ | $31.59_{\pm1.15}$ | $38.15_{\pm0.92}$ | $60.00_{\pm2.81}$ | $60.54_{\pm3.67}$ | $78.02_{\pm1.15}$ | $89.95_{\pm0.47}$ | $85.35_{\pm1.57}$ |
| PairNorm | $60.27_{\pm4.34}$ | $48.43_{\pm6.14}$ | $27.40_{\pm1.24}$ | $50.44_{\pm2.04}$ | $62.74_{\pm2.82}$ | $58.92_{\pm3.15}$ | $73.59_{\pm1.47}$ | $87.53_{\pm0.44}$ | $85.79_{\pm1.01}$ |
| GraphSAGE | $82.43_{\pm6.14}$ | $81.18_{\pm5.56}$ | $34.23_{\pm0.99}$ | $41.61_{\pm0.74}$ | $58.73_{\pm1.68}$ | $75.95_{\pm5.01}$ | $76.04_{\pm1.30}$ | $88.45_{\pm0.50}$ | $86.90_{\pm1.04}$ |
| GCN | $55.14_{\pm5.16}$ | $51.76_{\pm3.06}$ | $27.32_{\pm1.10}$ | $53.43_{\pm2.01}$ | $64.82_{\pm2.24}$ | $60.54_{\pm5.30}$ | $76.50_{\pm1.36}$ | $88.42_{\pm0.50}$ | $86.98_{\pm1.27}$ |
| GAT | $52.16_{\pm6.63}$ | $49.41_{\pm4.09}$ | $27.44_{\pm0.89}$ | $40.72_{\pm1.55}$ | $60.26_{\pm2.50}$ | $61.89_{\pm5.05}$ | $76.55_{\pm1.23}$ | $87.30_{\pm1.10}$ | $86.33_{\pm0.48}$ |
| MLP | $80.81_{\pm4.75}$ | $85.29_{\pm3.31}$ | $36.53_{\pm0.70}$ | $28.77_{\pm1.56}$ | $46.21_{\pm2.99}$ | $81.89_{\pm6.40}$ | $74.02_{\pm1.90}$ | $87.16_{\pm0.37}$ | $75.69_{\pm2.00}$ |

a sheaf at $t = 0$ as a function of $\mathbf{X}(0)$, and we keep the sheaf constant over time. For the first model, we learn a sheaf with general maps $\mathcal{F}_{v \trianglelefteq e} \in \mathbb{R}$. For the second model, we use a similar layer but constraint $\mathcal{F}_{v \trianglelefteq e} = \mathcal{F}_{u \trianglelefteq e}$, obtaining a weighted graph Laplacian.

Figure 5 presents the results across five seeds. As expected, for diffusion time zero (i.e. no diffusion), we see that a linear classifier cannot separate the classes. At later times, the diffusion process using symmetric maps cannot perfectly fit the data. In contrast, with the more general sheaf diffusion, as time increases and the signal approaches the harmonic space, the model gets better and the features become linearly separable. In the last subfigure, we take a closer look at the sheaf that the model learns in the time limit by plotting a histogram of all the transport (scalar) maps $\mathcal{F}_{v \trianglelefteq e}^{\top}\mathcal{F}_{u \trianglelefteq e}$. In accordance with Proposition 10, the model learns a negative transport map for all edges. This shows that the model manages to avoid oversmoothing (see Appendix F for an experiment with $d > 1$).

**Real-world experiments.** We test our models on multiple real-world datasets [47, 53, 57, 61, 66] with an edge homophily coefficient $h$ ranging from $h = 0.11$ (very heterophilic) to $h = 0.81$ (very homophilic). Therefore, they offer a view of how a model performs over this entire spectrum. We evaluate our models on the 10 fixed splits provided by Pei et al. [53] and report the mean accuracy and standard deviation. Each split contains $48\%/32\%/20\%$ of nodes per class for training, validation and testing, respectively. As baselines, we use an ample set of GNN models that can be placed in three categories: (1) classical: GCN [39], GAT [68], GraphSAGE [31]; (2) models specifically designed for heterophilic settings: GGCN [72], Geom-GCN [53], H2GCN [75], GPRGNN [17], FAGCN [7], MixHop [1]; (3) models addressing oversmoothing: GCNII [16], PairNorm [74]. All the results are taken from Yan et al. [72], except for FAGCN and MixHop, which come from Lingam et al. [41] and Zhu et al. [75], respectively. All of these were evaluated on the same set of splits as ours. In Appendix F we also include experiments with continuous GNN models.

**Results.** From Table 1 we see that our models are first in $5/6$ benchmarks with high heterophily ($h < 0.3$) and second-ranked on the remaining one (i.e. Chameleon). At the same time, NSD also shows strong performance on the homophilic graphs by being within approximately 1% of the top model. Overall, NSD models are among the top three models on $8/9$ datasets. The $O(d)$-bundle diffusion model performs best overall confirming the intuition that it can better avoid overfitting, while also transforming the vectors in sufficiently complex ways. We also remark on the strong performance of the model learning diagonals maps, despite the simpler functional form of the Laplacian.

## 7 Related Work, Discussion, and Conclusion

**Sheaf Neural Networks & Sheaf Learning.** Sheaf Neural Networks [32] with a *hand-crafted* sheaf Laplacian were originally introduced in a toy experimental setting. Since then, they have remained completely unexplored, and we hope this paper will fill this lacuna. In contrast to [32], we provide an ample theoretical analysis justifying the use of sheaves in Graph ML and study for the first time how sheaves can be *learned from data* using neural networks. Furthermore, we present the first successful application of Sheaf Neural Networks on real-world datasets. Hansen and Ghrist [33]

have also considered learning a sheaf Laplacian by minimising directly in matrix space a regularised Dirichlet energy metric. Different from their approach, we learn the sheaf as part of an end-to-end model and use an efficient parametrisation that is independent of the size of the graph.

Follow-up works have also experimented with inferring a connection Laplacian directly from data at pre-processing time [3], combining sheaves with attention [4], and designing models based on the wave equation on sheaves [65]. Besides the sheaf Laplacians employed in all these works and ours, one can also use higher-order sheaf (connection) Laplacians that operate on higher-order tensors. These were shown to encode important information about the underlying symmetries in the data [55], which hints at the powerful data properties that Sheaf Neural Networks could potentially extract from these operators.

**Heterophily and Oversmoothing.**   While good empirical designs jointly addressing these two problems have been proposed before [17, 72], Yan et al. [72] is the only other work connecting the two theoretically. Their analysis [72] is very different in terms of methods and assumptions and, therefore, their results are completely orthogonal. Concretely, the authors analyse the performance of linear SGCs [69] (i.e. GCN without nonlinearities) on random attributed graphs. In contrast, our analysis is not probabilistic, focuses on diffusion PDEs and also extends to GCNs in the non-linear regime. Furthermore, we employ a new set of mathematical tools from cellular sheaf theory, which brings a new language and new tools to analyse these problems. Perhaps the only commonality is that both works find evidence for the benefits of negatively signed edges in GNNs, although with different mathematical motivations. At the same time, other recent works [21, 42] have shown that GCNs with finite layers (typically one) can perform well in heterophilic graphs (including bipartite). This is in no contradiction with our results, which consider an *infinite time/layer* regime (i.e. not finite) and *perfect* linear separation (i.e. a model that cannot fit the data can still achieve high accuracy).

**Category Theory and GNNs.**   From the perspective of category theory [43], cellular sheaves are a *functor* from a *category* describing the incidence structure of the graph to a *category* describing the data living on top of the graph. Informally, this says that the vertices and edges are mapped to some type of data (e.g. vector spaces) and the incidence relations between vertices and edges are mapped to some type of relation between the assigned data (e.g. linear maps between the vector spaces). The generality provided by this perspective could be used to extend the models described in this work to more exotic types of data such as lattices and their associated sheaf Laplacians [25]. At the same time, our work echoes other recent efforts to place GNNs on a categorical foundation [19, 22].

**Message Passing Neural Networks.**   The layer from Equation 6 can be seen as a form of GNN-FiLM layer [11, 54], where each node learns a linear message function conditioned on the features of the neighbours. Such models have been recently shown to perform well empirically in heterophilic settings [52]. At the same time, the model bares an algorithmic resemblance to GAT [68]. For a central node $v$ and a neighbouring node $u$, GAT learns an attention coefficient $a_{vu}$, while our model learns a matrix given by the block $(v, u)$ of $\Delta_{\mathcal{F}}$. Finally, a message-passing procedure based on parallel transport has also been proposed by Haan et al. [30] in the context of geometric graphs (meshes). In the absence of a natural geometric structure on arbitrary graphs, in our case, the transport structure is learned from data end-to-end.

**Limitations and societal impact.**   One of the main limitations of our theoretical analysis is that it does not address the generalisation properties of sheaves, but this remains a major impediment for the entire field of deep learning. Nonetheless, our setting was sufficient to produce many valuable insights about heterophily and oversmoothing and a basic understanding of what various types of sheaves can and cannot do. Much more work remains to be done in this direction, and we expect to see further cross-fertilization between ML and algebraic topology in the future. Finally, due to the theoretical nature of this work, we do not foresee any immediate negative societal impacts.

**Conclusion.**   In this work, we used cellular sheaf theory to provide a novel topological perspective on heterophily and oversmoothing in GNNs. We showed that the underlying sheaf structure of the graph is intimately connected with both of these important factors affecting the performance of GNNs. To mitigate this, we proposed a new paradigm for graph representation learning where models not only evolve the features at each layer but also the underlying geometry of the graph. In practice, we demonstrated that this framework achieves competitive results in heterophilic settings.

## Acknowledgments and Disclosure of Funding

We are grateful to Iulia Duta, Dobrik Georgiev and Jacob Deasy for valuable comments on an earlier version of this manuscript. CB would also like to thank the Twitter Cortex team for making the research internship a fantastic experience. This research was supported in part by ERC Consolidator grant No. 724228 (LEMAN).

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
