# A Harmonic Space Proofs

**Proposition 3.** *If $\mathcal{F}$ is a discrete $O(d)$ bundle over a connected graph and $r := \max_{\gamma_{v\to u}, \gamma'_{v\to u}} ||\mathbf{P}^{\gamma}_{v\to u} - \mathbf{P}^{\gamma'}_{v\to u}||$, then we have $\lambda_0^{\mathcal{F}} \leq r^2/2$.*

*Proof.* We first note that on a discrete $O(d)$ bundle the degree operator $D_v = d_v \mathbf{I}$ since by orthogonality $\mathcal{F}^{\top}_{v\trianglelefteq e} \mathcal{F}_{v\trianglelefteq e} = \mathbf{I}$. We can use the Rayleigh quotient to characterize $\lambda_0^{\mathcal{F}}$ as

$$\lambda_0^{\mathcal{F}} = \min_{\mathbf{x}\in\mathbb{R}^{nd}} \frac{\langle \mathbf{x}, \Delta_{\mathcal{F}}\mathbf{x}\rangle}{||\mathbf{x}||^2}.$$

Fix $v \in V$ and choose a minimal path $\gamma_{v\to u}$ for all $u \in V$. For an arbitrary non-zero $\mathbf{z}_v$, consider the signal $\mathbf{z}_u = P^{\gamma}_{v\to u}\mathbf{z}_v$ and we set $\tilde{\mathbf{z}}_u \sqrt{d_u} = \mathbf{z}_u$.

$$||\mathcal{F}_{u\trianglelefteq e}\frac{\mathbf{z}_u}{\sqrt{d_u}} - \mathcal{F}_{w\trianglelefteq e}\frac{\mathbf{z}_w}{\sqrt{d_w}}||^2 = ||\tilde{\mathbf{z}}_u - (\mathcal{F}^{\top}_{u\trianglelefteq e}\mathcal{F}_{w\trianglelefteq e})\tilde{\mathbf{z}}_w||^2 = ||P^{\gamma}_{v\to u}\tilde{\mathbf{z}}_v - (\mathcal{F}^{\top}_{u\trianglelefteq e}\mathcal{F}_{w\trianglelefteq e})\mathbf{P}^{\gamma}_{v\to w}\tilde{\mathbf{z}}_v||^2,$$

where we have again used that the maps are orthogonal. Since $(\mathcal{F}^{\top}_{u\trianglelefteq e}\mathcal{F}_{w\trianglelefteq e})\mathbf{P}^{\gamma}_{v\to w} = \mathbf{P}^{\gamma'}_{v\to u}$ we find that the right hand side can be bound from above by $r^2||\tilde{\mathbf{z}}_v||^2$. Therefore, by using Definition 14 we finally obtain

$$\lambda_0^{\mathcal{F}} = \min_{\mathbf{x}\in R^{nd}} \frac{\langle \mathbf{x}, \Delta_{\mathcal{F}}\mathbf{x}\rangle}{||\mathbf{x}||^2} \leq \frac{\langle \mathbf{z}, \Delta_{\mathcal{F}}\mathbf{z}\rangle}{||\mathbf{z}||^2} = \frac{1}{2}\frac{\sum_{u\sim w}||\mathcal{F}_{u\trianglelefteq e}\frac{\mathbf{z}_u}{\sqrt{d_u}} - \mathcal{F}_{w\trianglelefteq e}\frac{\mathbf{z}_w}{\sqrt{d_w}}||^2}{||\mathbf{z}||^2} \leq \frac{r^2}{2}\frac{\sum_{u\sim w}||\tilde{\mathbf{z}}_v||^2}{||\mathbf{z}||^2}.$$

Since the transport maps are all orthogonal we get

$$||\mathbf{z}||^2 = \sum_{u} d_u ||\mathbf{P}^{\gamma}_{v\to u}\tilde{\mathbf{z}}_v||^2 = \sum_{u} d_u ||\tilde{\mathbf{z}}_v||^2 = \sum_{u\sim w}||\tilde{\mathbf{z}}_v||^2.$$

We conclude that

$$\lambda_0^{\mathcal{F}} \leq \frac{r^2}{2}\frac{\sum_{u\sim w}||\tilde{\mathbf{z}}_v||^2}{||\mathbf{z}||^2} = \frac{r^2}{2}.$$

$\square$

**Proposition 4.** *If $\mathcal{F}$ is a discrete $O(d)$ bundle over a connected graph and $\mathbf{x} \in H^0(G, \mathcal{F})$, then for any cycle $\gamma$ based at $v \in V$ we have $\mathbf{x}_v \in \ker(\mathbf{P}^{\gamma}_{v\to v} - \mathbf{I})$.*

*Proof.* Assume that $\mathbf{x} \in H^0(G, \mathcal{F})$ and consider $v \in V$ and any cycle based at $v$ denoted by $\gamma_{v\to v} = (v_0 = v, v_1, \ldots, v_L = v)$. According to the Hodge Theorem we have that

$$\mathcal{F}_{v_{i+1}\trianglelefteq e}\mathbf{x}_{v_{i+1}} = \mathcal{F}_{v_i\trianglelefteq e}\mathbf{x}_{v_i} \implies \mathbf{x}_{v_{i+1}} = (\mathcal{F}^{\top}_{v_{i+1}}\mathcal{F}_{v_i})\mathbf{x}_{v_i} := \rho_{v_i\to v_{i+1}}\mathbf{x}_{v_i}.$$

By composing all the maps we find:

$$\mathbf{x}_v = \rho_{v_{L-1}\to v_L}\cdots\rho_{v_0\to v_1}x_v = \mathbf{P}^{\gamma}_{v\to v}\mathbf{x}_v$$

which completes the proof. $\square$

**Proposition 5.** *Let $\mathcal{F}$ be a discrete $O(d)$ bundle over a connected graph $G$ with $n$ nodes and let $||(\mathbf{P}^{\gamma}_{v\to v} - \mathbf{I})\mathbf{x}_v|| \geq \epsilon||\mathbf{x}_v||$ for all cycles $\gamma_{v\to v}$. Then $\lambda_0^{\mathcal{F}} \geq \epsilon^2(2\mathrm{diam}(G)n\,d_{max})^{-1}$.*

*Proof.* If $\epsilon = 0$ there is nothing to prove. Assume that $\epsilon > 0$. By Proposition 4 we derive that the harmonic space is trivial and hence $\lambda_0^{\mathcal{F}} > 0$. Consider a *unit* eigenvector $\mathbf{x} \in \ker(\Delta_{\mathcal{F}} - \lambda_0^{\mathcal{F}}I)$ and let $v \in V$ such that $||\mathbf{x}_v|| \geq ||\mathbf{x}_u||$ for $u \neq v$. There exists a cycle $\gamma$ based at $v$ such that $\mathbf{P}^{\gamma}_v\mathbf{x}_v \neq \mathbf{x}_v$ for otherwise we could extend $\mathbf{x}_v \neq 0$ to any other node independently of the path choice and hence find a non-trivial harmonic signal. In particular, we can assume this cycle to be non-degenerate, otherwise if there existed a non-trivial degenerate loop contained in $\gamma$ that does not fix $\mathbf{x}$ we could consider this loop instead of $\gamma$ for our argument. Let us write this path as $(v_0 = v, v_1, \ldots, v_L = v)$ and consider the rescaled signal $\tilde{\mathbf{x}}_v\sqrt{d_v} = \mathbf{x}_v$. By assumption we have

$$\epsilon||\tilde{\mathbf{x}}_v|| \leq ||(\mathbf{P}^{\gamma}_{v\to v} - \mathbf{I})\tilde{\mathbf{x}}_v|| = ||(\rho_{v_{L-1}\to v_L}\cdots\rho_{v_0\to v_1} - \mathbf{I})\tilde{\mathbf{x}}_v||$$

$$= ||\mathcal{F}_{v_{L-1}}\rho_{v_{L-2}\to v_{L-1}}\cdots\rho_{v_0\to v_1}\tilde{\mathbf{x}}_v - \mathcal{F}_{v_L=v}\tilde{\mathbf{x}}_v||$$

$$= ||\mathcal{F}_{v_{L-1}}\rho_{v_{L-2}\to v_{L-1}}\cdots\rho_{v_0\to v_1}\tilde{\mathbf{x}}_v - \mathcal{F}_{v_{L-1}}\tilde{\mathbf{x}}_{v_{L-1}} + \mathcal{F}_{v_{L-1}}\tilde{\mathbf{x}}_{v_{L-1}} - \mathcal{F}_{v_L=v}\tilde{\mathbf{x}}_v||$$

$$\leq ||\rho_{v_{L-2}\to v_{L-1}}\cdots\rho_{v_0\to v_1}\tilde{\mathbf{x}}_v - \tilde{\mathbf{x}}_{v_{L-1}}|| + ||\mathcal{F}_{v_{L-1}}\tilde{\mathbf{x}}_{v_{L-1}} - \mathcal{F}_{v_L=v}\tilde{\mathbf{x}}_v||.$$

By iterating the approach above we find:

$$\epsilon||\tilde{\mathbf{x}}_v|| \leq \sum_{i=0}^{L}||\mathcal{F}_{v_i}\tilde{\mathbf{x}}_{v_i} - \mathcal{F}_{v_{i+1}}\tilde{\mathbf{x}}_{v_{i+1}}|| \leq \sqrt{L}\left(\sum_{i=0}^{L}||\mathcal{F}_{v_i}\tilde{\mathbf{x}}_{v_i} - \mathcal{F}_{v_{i+1}}\tilde{\mathbf{x}}_{v_{i+1}}||^2\right)^{\frac{1}{2}}$$

$$= \sqrt{L}\left(\sum_{i=0}^{L}||\mathcal{F}_{v_i}\frac{\mathbf{x}_{v_i}}{\sqrt{d_{v_i}}} - \mathcal{F}_{v_{i+1}}\frac{\mathbf{x}_{v_{i+1}}}{\sqrt{d_{v_{i+1}}}}||^2\right)^{\frac{1}{2}}.$$

From Definition 14 we derive that the last term can be bounded from above by $\sqrt{2LE_\mathcal{F}(\mathbf{x})} = \sqrt{2L\langle\mathbf{x}, \Delta_\mathcal{F}\mathbf{x}\rangle}$. Therefore, we conclude:

$$\epsilon\frac{||\mathbf{x}_v||}{\sqrt{d_v}} \leq \sqrt{2L\langle\mathbf{x}, \Delta_\mathcal{F}\mathbf{x}\rangle} = \sqrt{2L\lambda_0^\mathcal{F}}||\mathbf{x}|| \leq 2\sqrt{\mathrm{diam}(G)\lambda_0^\mathcal{F}}.$$

By construction we get $||\mathbf{x}_v|| \geq 1/\sqrt{n}$, meaning that

$$\lambda_0^\mathcal{F} \geq \frac{\epsilon^2}{2\mathrm{diam}(G)}\frac{1}{n\,d_{\max}}.$$

$\square$

**Lemma 6.** *Let $\mathcal{F}$ be a discrete $O(d)$ bundle over a connected graph $G$. Then $\dim(H^0) \leq d$ and $\dim(H^0) = d$ if and only if the transport is path-independent.*

*Proof.* We first note that the argument below extends to weighted $O(d)$-bundles as well. Let $\mathbf{x} \in H^0(G, \mathcal{F})$. According to Proposition 4, given $v, u \in V$, we see that $x_u = \mathbf{P}_{v\to u}^\gamma x_v$ for any path $\gamma_{v\to u}$. It means that the harmonic space is uniquely determined by the choice of $\mathbf{x}_v \in \mathcal{F}(v)$. Explicitly, given any cycle $\gamma$ based at $v$, we know that $\mathbf{x}_v \in \ker(\mathbf{P}_{v\to v}^\gamma - I)$. If the transport is everywhere path-independent, then the kernel coincides with the whole stalk $\mathcal{F}(v)$ and hence we can extend any basis $\{\mathbf{x}_{v_i}\} \in \mathcal{F}(v) \cong \mathbb{R}^d$ to a basis in $H^0(G, \mathcal{F})$ via the transport maps, i.e. $\dim(H^0(G, \mathcal{F})) = d$. If instead there exists a transport map over a cycle $\gamma_{v\to v}$ with non-trivial fixed points, then $\ker(\mathbf{P}_{v\to v}^\gamma - I) < \mathcal{F}(v) \cong \mathbb{R}^d$ and hence $\dim(H^0(G, \mathcal{F})) < d$. $\square$

## B  Proofs for the Power of Sheaf Diffusion

**Definition 19.** *Let $G = (V, \mathbf{W})$ be a weighted graph, where $\mathbf{W}$ is a matrix with $w_{vu} = w_{uv} \geq 0$ for all $v \neq u \in V$, $w_{vv} = 0$ for all $v \in V$, and $(v, u)$ is an edge if and only if $w_{vu} > 0$.*

The graph Laplacian of a weighted graph is $\mathbf{L} = \mathbf{D} - \mathbf{W}$, where $\mathbf{D}$ is the diagonal matrix of weighted degrees (i.e. $d_v = \sum_u w_{vu}$). Its normalised version is $\widetilde{L} = \mathbf{D}^{-1/2}\mathbf{L}\mathbf{D}^{-1/2}$.

**Proposition 20.** *Let $G$ be a graph. The set $\{\Delta_\mathcal{F} \mid (G, \mathcal{F}) \in \mathcal{H}_{\mathrm{sym}}^1\}$ is isomorphic to the set of all possible weighted graph Laplacians over $G$.*

*Proof.* We prove only one direction. Let $\mathbf{W}$ be a choice of valid weight matrix for the graph $G$. We can construct a sheaf $(G, \mathcal{F}) \in \mathcal{H}_{\mathrm{sym}}^1$ such that for all edges $v, u \trianglelefteq e$ we have that $\mathcal{F}_{v\trianglelefteq e} = \mathcal{F}_{u\trianglelefteq e} = \pm\sqrt{w_{vu}}$. Then, $\mathcal{L}_{vu} = -w_{vu}$ and $\mathcal{L}_{vv} = \sum_e||\mathcal{F}_{v\trianglelefteq e}||^2 = \sum_u w_{vu}$. The equality for the normalised version of the Laplacians follows directly. $\square$

We state the following Lemma without proof based on Theorem 3.1 in Hansen and Ghrist [35].

**Lemma 21.** *Solutions $\mathbf{X}(t)$ to the diffusion in Equation 3 converge as $t \to \infty$ to the orthogonal projection of $\mathbf{X}(0)$ onto $\ker(\Delta_\mathcal{F})$.*

Due to this Lemma, the proofs below rely entirely on the structure of $\ker(\Delta_\mathcal{F})$ that one obtains for certain $(G, \mathcal{F})$.

**Proposition 8.** *Let $\mathcal{G}$ be the set of connected graphs $G = (V, E)$ with two classes $A, B \subset V$ such that for each $v \in A$, there exists $u \in A$ and an edge $(v, u) \in E$. Then $\mathcal{H}_{\mathrm{sym}}^1$ has linear separation power over $\mathcal{G}$.*

*Proof.* Let $G = (V, E)$ be a graph with two classes $A, B \subset V$ such that for each $v \in A$, there exists $u \in A$ and an edge $(v, u) \in E$. Additionally, let $\mathbf{x}(0)$ be any channel of the feature matrix $\mathbf{X}(0) \in \mathbb{R}^{n \times f}$.

We can construct a sheaf $(\mathcal{F}, G) \in \mathcal{H}^1_{\text{sym}}$ as follows. For all nodes $v \in V$ and edges $e \in E$, $\mathcal{F}(v) \cong \mathcal{F}(e) \cong \mathbb{R}$. For all $v, u \in A$ and edge $(u, v) \in E$, set $\mathcal{F}_{v \trianglelefteq e} = \mathcal{F}_{u \trianglelefteq e} = \sqrt{\alpha} > 0$. Otherwise, set $\mathcal{F}_{v \trianglelefteq e} = 1$.

Denote by $h_v$ the number of neighbours of node $v$ in the same class as $v$. Note that based on the assumptions, $h_v > 1$ if $v \in A$. Then the only harmonic eigenvector of $\Delta_{\mathcal{F}}$ is:

$$\mathbf{a}_v = \begin{cases} \sqrt{d_v + h_v(\alpha - 1)}, & v \in A \\ \sqrt{d_v}, & v \in B \end{cases} \tag{7}$$

Denote its unit-normalised version $\widetilde{\mathbf{a}} := \frac{\mathbf{a}}{\|\mathbf{a}\|}$. In the limit of the diffusion process, the features converge to $\mathbf{h} = \langle \mathbf{x}(0), \widetilde{\mathbf{a}} \rangle \widetilde{\mathbf{a}}$ by Lemma 21. Assuming, $\mathbf{x}(0) \notin \ker(\Delta_{\mathcal{F}})^\perp$, which is nowhere dense in $\mathbb{R}^n$ and, without loss of generality, that $\langle \mathbf{x}(0), \widetilde{\mathbf{a}} \rangle > 0$, for sufficiently large $\alpha$, $\widetilde{\mathbf{a}}_v \geq \widetilde{\mathbf{a}}_u$ for all $v \in A, u \in B$. $\qquad \square$

**Proposition 9.** *Let $\mathcal{G}$ be the set of connected bipartite graphs $G = (A, B, E)$, with partitions $A, B$ forming two classes and $|A| = |B|$. Then $\mathcal{H}^1_{\text{sym}}$ cannot linearly separate the classes of any graph in $\mathcal{G}$ for any initial conditions $\mathbf{X}(0) \in \mathbb{R}^{n \times f}$.*

*Proof.* Let $G = (A, B, E)$ be a bipartite graph with $|A| = |B|$ and let $\mathbf{x}(0) \in \mathbb{R}^n$ be any channel of the feature matrix $\mathbf{X}(0) \in \mathbb{R}^{n \times f}$.

Consider an arbitrary sheaf $(G, \mathcal{F}) \in \mathcal{H}^1_{\text{sym}}$. Since the graph is connected, the only harmonic eigenvector of $\Delta_{\mathcal{F}}$ is $\mathbf{y} \in \mathbb{R}^n$ with $\mathbf{y}_v = \sqrt{\sum_{v \trianglelefteq e} \|\mathcal{F}_{v \trianglelefteq e}\|^2}$ (i.e. the square root of the weighted degree). Based on Lemma 21, the diffusion process converges in the limit (up to a scaling) to $\langle \mathbf{x}, \mathbf{y} \rangle \mathbf{y}$. For the features to be linearly separable we require that $\langle \mathbf{x}, \mathbf{y} \rangle \neq 0$ and, without loss of generality, for all $v \in A, u \in B$ that $\mathbf{y}_v < \mathbf{y}_u \Leftrightarrow \sum_{v \trianglelefteq e} \|\mathcal{F}_{v \trianglelefteq e}\|^2 < \sum_{u \trianglelefteq e} \|\mathcal{F}_{u \trianglelefteq e}\|^2$.

Suppose for the sake of contradiction there exists a sheaf in $\mathcal{H}^1_{\text{sym}}$ with such a harmonic eigenvector. Then, because $|A| = |B|$:

$$\sum_{v \in A} \sum_{v \trianglelefteq e} \|\mathcal{F}_{v \trianglelefteq e}\|^2 < \sum_{u \in B} \sum_{u \trianglelefteq e} \|\mathcal{F}_{u \trianglelefteq e}\|^2 \Leftrightarrow \sum_{v \in A} \sum_{v \trianglelefteq e} \|\mathcal{F}_{v \trianglelefteq e}\|^2 - \sum_{u \in B} \sum_{u \trianglelefteq e} \|\mathcal{F}_{u \trianglelefteq e}\|^2 < 0$$

$$\Leftrightarrow \sum_{e \in E} \|\mathcal{F}_{v \trianglelefteq e}\|^2 - \|\mathcal{F}_{u \trianglelefteq e}\|^2 < 0$$

However, because $(\mathcal{F}, G) \in \mathcal{H}^1_{\text{sym}}$, we have $\mathcal{F}_{v \trianglelefteq e} = \mathcal{F}_{u \trianglelefteq e}$ and the sum above is zero. $\qquad \square$

**Proposition 10.** *Let $\mathcal{G}$ contain all the connected graphs $G = (V, E)$ with two classes $A, B \subseteq V$. Consider a sheaf $(\mathcal{F}; G) \in \mathcal{H}^1$ with $\mathcal{F}_{v \trianglelefteq e} = -\alpha_e$ if $v \in A$ and $\mathcal{F}_{u \trianglelefteq e} = \alpha_e$ if $u \in B$ with $\alpha_e > 0$ for all $e \in E$. Then the diffusion induced by $(\mathcal{F}; G)$ can linearly separate the classes of $G$ for almost all initial conditions, and $\mathcal{H}^1$ has linear separation power over $\mathcal{G}$.*

*Proof.* Let $G = (V, E)$ be a connected graph with two classes $A, B \subset V$. Additionally, let $\mathbf{x}(0)$ be any channel of the feature matrix $\mathbf{X}(0) \in \mathbb{R}^{n \times f}$. Any sheaf of the described type has a single harmonic eigenvector by virtue of Lemma 6, and it has the form:

$$\mathbf{y}_v = \begin{cases} +\sqrt{\sum_{v \trianglelefteq e} \alpha_e}, & v \in A \\ -\sqrt{\sum_{v \trianglelefteq e} \alpha_e}, & v \in B \end{cases} \tag{8}$$

Assume $\mathbf{x}(0) \notin \ker(\Delta_{\mathcal{F}})^\perp$, which is nowhere dense in $\mathbb{R}^n$ and, without loss of generality, that $\langle \mathbf{x}(0), \mathbf{y} \rangle > 0$. Then, $\mathbf{y}_v > 0 > \mathbf{y}_u$ for all $v \in A, u \in B$. $\qquad \square$

Next, we showed that using signed relations is necessary in $d = 1$ and simply using positive asymmetric relations is not sufficient in this dimension.

**Definition 22.** *The class of sheaves over $G$ with non-zero maps, one-dimensional stalks, and similarly signed restriction maps $\mathcal{H}_+^1 := \{(\mathcal{F}, G) \mid \mathcal{F}_{v \trianglelefteq e} \mathcal{F}_{u \trianglelefteq e} > 0\}$*

**Proposition 23.** *Let $G$ be the connected graph with two nodes belonging to two different classes. Then $\mathcal{H}_+^1$ cannot linearly separate the two nodes for any initial conditions $\mathbf{X} \in \mathbb{R}^{2 \times f}$.*

*Proof.* Let $G$ be the connected graph with two nodes $V = \{v, u\}$. Then any sheaf $(\mathcal{F}, G) \in \mathcal{H}_+^1(G)$ has restriction maps of the form $\mathcal{F}_{v \trianglelefteq e} = \alpha$, $\mathcal{F}_{u \trianglelefteq e} = \beta$ and (without loss of generality) $\alpha, \beta > 0$. As before, the only (unnormalized) harmonic eigenvector for a sheaf of this form is $\mathbf{y} = (|\alpha|\beta|, \alpha|\beta|) = (\alpha\beta, \alpha\beta)$. Since this is a constant vector, the two nodes are not separable in the diffusion limit. $\square$

We state the following result without a proof (see Exercise 4.1 in Bishop [6]).

**Lemma 24.** *Let $A$ and $B$ be two sets of points in $\mathbb{R}^n$. If their convex hulls intersect, the two sets of points cannot be linearly separable.*

**Proposition 11.** *Let $G$ be a connected graph with $C \geq 3$ classes. Then, $\mathcal{H}^1$ cannot linearly separate the classes of $G$ for any initial conditions $\mathbf{X}(0) \in \mathbb{R}^{n \times f}$.*

*Proof.* If the sheaf has a trivial global section, all features converge to zero in the diffusion limit. Suppose $H^0(G, \mathcal{F})$ is non-trivial. Since $G$ is connected and all the restriction maps are invertible, by Lemma 6, $\dim(H^0) = 1$.

In that case, let $\mathbf{h}$ be the unit-normalised harmonic eigenvector of $\Delta_{\mathcal{F}}$. By Lemma 21, for any node $v$, its scalar feature in channel $k \leq f$ is given by $x_v^k(\infty) = \langle \mathbf{x}^k(0), \mathbf{h} \rangle \mathbf{h}_v$. Note that we can always find three nodes $v, u, w$ belonging to three different classes such that $\mathbf{h}_v \leq \mathbf{h}_u \leq \mathbf{h}_v$. Then, there exists a convex combination $\mathbf{h}_u = \alpha \mathbf{h}_v + (1 - \alpha)\mathbf{h}_w$, with $\alpha \in [0, 1]$. Therefore:

$$\mathbf{x}_u^k(\infty) = \langle \mathbf{x}^k(0), \mathbf{h} \rangle \mathbf{h}_u = \alpha \langle \mathbf{x}^k(0), \mathbf{h} \rangle \mathbf{h}_v + (1 - \alpha) \langle \mathbf{x}^k(0), \mathbf{h} \rangle \mathbf{h}_w = \alpha \mathbf{x}_v^k(\infty) + (1 - \alpha)\mathbf{x}_w^k(\infty). \tag{9}$$

Since this is true for all channels $k \leq f$, it follows that $\mathbf{x}_u(\infty) = \alpha \mathbf{x}_v(\infty) + (1 - \alpha)\mathbf{x}_w(\infty)$. Because $\mathbf{x}_u(\infty)$ is in the convex hull of the points belonging to other classes, by Lemma 24, the class of $v$ is not linearly separable from the other classes. $\square$

**Proposition 12.** *Let $\mathcal{G}$ be the set of connected graphs with nodes belonging to $C \geq 3$ classes. Then for $d \geq C$, $\mathcal{H}_{\text{diag}}^d$ has linear separation power over $\mathcal{G}$.*

*Proof.* Let $G = (V, E)$ be a connected graph with $C$ classes and $(\mathcal{F}, G)$, an arbitrary sheaf in $\mathcal{H}_{\text{diag}}^d$. Because $\mathcal{F}$ has diagonal restriction maps, there is no interaction during diffusion between the different dimensions of the stalks. Therefore, the diffusion process can be written as $d$ independent diffusion processes, where the $i$-th process uses a sheaf $\mathcal{F}^i$ with all stalks isomorphic to $\mathbb{R}$ and $\mathcal{F}_{v \trianglelefteq e}^i = \mathcal{F}_{v \trianglelefteq e}(i, i)$ for all $v \in V$ and incident edges $e$. Therefore, we can construct $d$ sheaves $\mathcal{F}^i \in \mathcal{H}^1(G)$ with $i < d$ as in Proposition 10, where (in one vs all fashion) the two classes are given by the nodes in class $i$ and the nodes belonging to the other classes.

It remains to restrict that the projection of $\mathbf{x}(0)$ on any of the harmonic eigenvectors of $\Delta_{\mathcal{F}}$ in the standard basis is non-zero. Formally, we require $\mathbf{x}^i(0) \notin \ker(\Delta_{\mathcal{F}^i})^\perp$ for all positive integers $i \leq d$. Since $\ker(\Delta_{\mathcal{F}^i})^\perp$ is nowhere dense in $\mathbb{R}^n$, $\mathbf{x}(0)$ belongs to the direct sum of dense subspaces, which is dense. $\square$

**Lemma 25.** *Let $G = (V, E)$ be a graph and $(\mathcal{F}, G)$ a (weighted) orthogonal vector bundle over $G$ with path-independent parallel transport and edge weights $\alpha_e$. Consider an arbitrary node $v^* \in V$ and denote by $\mathbf{e}_i$ the $i$-th standard basis vector of $\mathbb{R}^d$. Then $\{\mathbf{h}^1, \dots, \mathbf{h}^d\}$ form an orthogonal eigenbasis for the harmonic space of $\Delta_{\mathcal{F}}$, where:*

$$\mathbf{h}_v^i = \begin{cases} \mathbf{e}_i \sqrt{d_v^{\mathcal{F}}} \\ \mathbf{P}_{v \to w} \mathbf{e}_i \sqrt{d_v^{\mathcal{F}}} \end{cases} = \begin{cases} \mathbf{e}_i \sqrt{\sum_{v \trianglelefteq e} \alpha_e^2}, & v = v^* \\ \mathbf{P}_{v* \to w} \mathbf{e}_i \sqrt{\sum_{v \trianglelefteq e} \alpha_e^2}, & otherwise \end{cases} \tag{10}$$

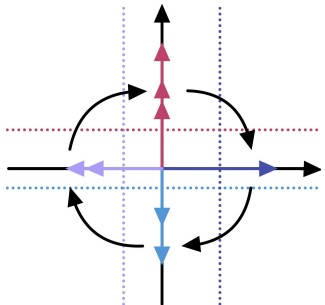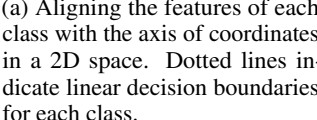
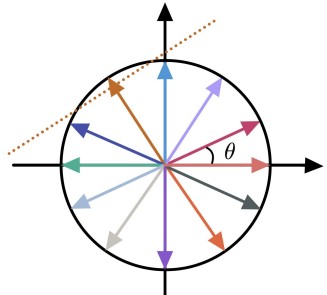

(a) Aligning the features of each class with the axis of coordinates in a 2D space. Dotted lines indicate linear decision boundaries for each class.

(b) Separating an arbitrary number of classes when the graph is regular. Dotted line shows an example decision boundary for one of the classes.

Figure 6: Proof sketch for Lemma 27 and Proposition 28.

*Proof.* First, we show that $\mathbf{h}_v^i$ is harmonic.

$$E_{\mathcal{F}}(\mathbf{h}_v^i) = \frac{1}{2} \sum_{v,u,e:=(v,u)} \| \frac{1}{\sqrt{d_v^{\mathcal{F}}}} \mathcal{F}_{v \leq e} \mathbf{h}_v - \frac{1}{\sqrt{d_u^{\mathcal{F}}}} \mathcal{F}_{u \trianglelefteq e} \mathbf{h}_u \|_2^2 \tag{11}$$

$$= \frac{1}{2} \sum_{v,u,e:=(v,u)} \| \mathcal{F}_{v \leq e} \mathbf{P}_{v^* \to v} \mathbf{e}_i - \mathcal{F}_{u \trianglelefteq e} \mathbf{P}_{v^* \to u} \mathbf{e}_i \|_2^2 \tag{12}$$

$$= \frac{1}{2} \sum_{v,u,e:=(v,u)} \| \mathcal{F}_{v \leq e} \mathbf{P}_{u \to v} \mathbf{P}_{v^* \to u} \mathbf{e}_i - \mathcal{F}_{u \trianglelefteq e} \mathbf{P}_{v^* \to u} \mathbf{e}_i \|_2^2 \quad \text{By path independence}$$
$$\tag{13}$$

$$= \frac{1}{2} \sum_{v,u,e:=(v,u)} \| \mathcal{F}_{v \leq e} \mathcal{F}_{v \trianglelefteq e}^\top \mathcal{F}_{u \trianglelefteq e} \mathbf{P}_{v^* \to u} \mathbf{e}_i - \mathcal{F}_{u \trianglelefteq e} \mathbf{P}_{v^* \to u} \mathbf{e}_i \|_2^2 \quad \text{By definition of } \mathbf{P}_{u \to v}$$
$$\tag{14}$$

$$= \frac{1}{2} \sum_{v,u,e:=(v,u)} \| \mathcal{F}_{u \trianglelefteq e} \mathbf{P}_{v^* \to u} \mathbf{e}_i - \mathcal{F}_{u \trianglelefteq e} \mathbf{P}_{v^* \to u} \mathbf{e}_i \|_2^2 = 0 \quad \text{Orthogonality of } \mathcal{F}_{v \trianglelefteq e}$$
$$\tag{15}$$

For orthogonality, notice that for any $i, j \leq d$ and $v \in V$, it holds that:

$$\langle \mathbf{h}_v^i, \mathbf{h}_v^j \rangle = \langle \mathbf{P}_{v* \to w} \mathbf{e}_i \sqrt{d_v^{\mathcal{F}}}, \mathbf{P}_{v* \to w} \mathbf{e}_j \sqrt{d_v^{\mathcal{F}}} \rangle = \sqrt{d_v^{\mathcal{F}}} \sqrt{d_v^{\mathcal{F}}} \langle \mathbf{e}_i, \mathbf{e}_j \rangle = 0 \tag{16}$$

$\square$

**Lemma 26.** *Let* $\mathbf{R}_1, \mathbf{R}_2$ *be two 2D rotation matrices and* $\mathbf{e}_1, \mathbf{e}_2$ *the two standard basis vectors of* $\mathbb{R}^2$. *Then* $\langle \mathbf{R}_1 \mathbf{e}_1, \mathbf{R}_2 \mathbf{e}_2 \rangle = -\langle \mathbf{R}_1 \mathbf{e}_2, \mathbf{R}_2 \mathbf{e}_1 \rangle$.

*Proof.* The angle between $\mathbf{e}_1$ and $\mathbf{e}_2$ is $\frac{\pi}{2}$. Letting $\phi, \theta$ be the positive rotation angles of the two matrices, the first inner product is equal to $\cos(\pi/2 + (\phi - \theta))$ while the second is $\cos(\pi/2 - (\phi - \theta))$. The result follows from applying the trigonometric identity $\cos(\pi/2 + x) = -\sin x$. $\square$

We first prove Theorem 13 in dimension two in the following lemma and then we will look at the general case.

**Lemma 27.** *Let* $\mathcal{G}$ *be the class of connected graphs with* $C \leq 4$ *classes. Then,* $\mathcal{H}_{\text{orth}}^2(G)$ *has linear separation power over* $\mathcal{G}$.

*Proof.* Idea: We can use rotation matrices to align the harmonic features of the classes with the axis of coordinates as in Figure 6a. Then, for each side of each axis, we can find a separating hyperplane separating each class from all the others.

Let $G$ be a connected graph with $C \le 4$ classes. Denote by $\mathcal{P}$ the following set of rotation matrices together with their signed-flipped counterparts:

$$\mathbf{R}_1 = \begin{bmatrix} 1 & 0 \\ 0 & 1 \end{bmatrix}, \quad \mathbf{R}_2 = \begin{bmatrix} 0 & -1 \\ 1 & 0 \end{bmatrix} \tag{17}$$

and by $\mathcal{C} = \{1, \dots, C\}$ the set of all class labels. Then, fix a node $v^* \in V$ and construct an injective map $g : C \to \mathcal{P}$ assigning each class label one of the signed basis vectors such that $g(c(v^*)) = \mathbf{R}_1$, where $c(v^*)$ denotes the class of node $v^*$.

Then, we can construct a sheaf $(G, \mathcal{F}) \in \mathcal{H}^2_{\mathrm{orth}}(G)$ in terms of certain parallel transport maps along each edge, that will depend on $\mathcal{P}$. For all nodes $v$ and edges $e$, $\mathcal{F}(v) \cong \mathcal{F}(e) \cong \mathbb{R}^2$. For each $u \in V$, we set $\mathbf{P}_{v^* \to u} = g(c(u))$. Then for all $v, u \in V$, set $\mathbf{P}_{v \to u} = \mathbf{P}_{v^* \to v} \mathbf{P}_{u \to v^*}^{-1}$. It is easy to see that the resulting parallel transport is path-independent because it depends purely on the classes of the endpoints of the path.

Based on Lemma 25, the $i$-th eigenvector of $\Delta_{\mathcal{F}}$ is $\mathbf{h}^i \in \mathbb{R}^{2 \times n}$ with $\mathbf{h}^i_u = \mathbf{P}_{v^* \to u} \mathbf{e}_i \sqrt{d_u}$. Now we will show that the projection of $\mathbf{x}(0)$ in this subspace will have a configuration as in Figure 6a up to a rotation.

Let $u, w$ be two nodes belonging to two different classes. Denote by $\alpha_i = \langle \mathbf{x}(0), \mathbf{h}^i \rangle$. Then the inner product between the features of nodes $u, w$ in the limit of the diffusion process is:

$$\langle \mathbf{P}_{v^* \to u} \sum_i \alpha_i \mathbf{e}_i \sqrt{d_u}, \mathbf{P}_{v^* \to w} \sum_j \alpha_j \mathbf{e}_j \sqrt{d_w} \rangle =$$

$$= \sqrt{d_u d_w} \Big[ \sum_{i \ne j} \alpha_i \alpha_j \langle \mathbf{P}_{v^* \to u} \mathbf{e}_i, \mathbf{P}_{v^* \to w} \mathbf{e}_j \rangle + \sum_k \alpha_k^2 \langle \mathbf{P}_{v^* \to u} \mathbf{e}_k, \mathbf{P}_{v^* \to w} \mathbf{e}_k \rangle \Big]$$

$$= \sqrt{d_u d_w} \Big[ \sum_{i < j} \alpha_i \alpha_j \big( \langle \mathbf{P}_{v^* \to u} \mathbf{e}_i, \mathbf{P}_{v^* \to w} \mathbf{e}_j \rangle + \langle \mathbf{P}_{v^* \to u} \mathbf{e}_j, \mathbf{P}_{v^* \to w} \mathbf{e}_i \rangle \big)$$

$$+ \sum_k \alpha_k^2 \langle \mathbf{P}_{v^* \to u} \mathbf{e}_k, \mathbf{P}_{v^* \to w} \mathbf{e}_k \rangle \Big] \tag{18}$$

$$= \sum_k \alpha_k^2 \langle \mathbf{P}_{v^* \to u} \mathbf{e}_k, \mathbf{P}_{v^* \to w} \mathbf{e}_k \rangle \qquad \text{(by Lemma 26)}$$

It can be checked that by substituting the transport maps $\mathbf{P}_{v^* \to u}, \mathbf{P}_{v^* \to w}$ with any $\mathbf{R}_a, \mathbf{R}_b$ from $\mathcal{P}$ such that $\mathbf{R}_a \ne \pm \mathbf{R}_b$, the inner product above is zero. Similarly, substituting any $\mathbf{R}_a = -\mathbf{R}_b$, the inner product is $-\sqrt{d_u d_w} \sum_k \alpha_k^2 = -\sqrt{d_u d_w} \|\mathbf{x}(0)\|^2$, which is equal to the product of the norms of the two vectors. Therefore, the diffused features of different classes are positioned at $\frac{\pi}{2}, \pi, \frac{3\pi}{2}$ from each other, as in Figure 6a. $\qquad \square$

**Proposition 13.** *Let $\mathcal{G}$ be the class of connected graphs with $C \le 2d$ classes. Then, for all $d \in \{2, 4\}$, $\mathcal{H}^d_{\mathrm{orth}}$ has linear separation power over $\mathcal{G}$.*

*Proof.* To generalise the proof in Lemma 27, we need to find a set $\mathcal{P}$ of size $d$ containing rotation matrices that make the projected features of different classes be pairwise orthogonal for any projection coefficients $\alpha$. For that, each term in Equation 18 must be zero for any coefficients $\alpha$.

Therefore, $\mathcal{P} = \{\mathbf{P}_0, \dots, \mathbf{P}_{d-1}\}$ must satisfy the following requirements:

1. $\mathbf{P}_0 = \mathbf{I} \in \mathcal{P}$, since transport for neighbours in the same class must be the identity. Therefore, $\mathbf{P}_0 \mathbf{P}_k = \mathbf{P}_k \mathbf{P}_0 = \mathbf{P}_k$ for all $k$.

2. Since $\langle \mathbf{P}_0 \mathbf{e}_i, \mathbf{P}_k \mathbf{e}_i \rangle = 0$ for all $i$ and $k \ne 0$, it follows that the diagonal elements of $\mathbf{P}_k$ are zero.

3. From $\langle \mathbf{P}_0 \mathbf{e}_i, \mathbf{P}_k \mathbf{e}_j \rangle = -\langle \mathbf{P}_0 \mathbf{e}_j, \mathbf{P}_k \mathbf{e}_i \rangle$ for all $i \ne j, k \ne 0$ and point (2) it follows that $\mathbf{P}_k^{-1} = \mathbf{P}_k^\top = -\mathbf{P}_k$. Therefore, $\mathbf{P}_k \mathbf{P}_k = -\mathbf{I}$ for all $k \ne 0$.

4. We have $\langle \mathbf{P}_k \mathbf{e}_i, \mathbf{P}_l \mathbf{e}_i \rangle = 0$ for all $i$ and $k \ne l$. Together with (3), it follows that the diagonal elements of $\mathbf{P}_k \mathbf{P}_l$ are zero.

5. We have $\langle \mathbf{P}_k \mathbf{e}_i, \mathbf{P}_l \mathbf{e}_j \rangle = -\langle \mathbf{P}_k \mathbf{e}_j, \mathbf{P}_l \mathbf{e}_i \rangle$ for all $i \neq j$, and $k \neq l$, with $k, l \neq 0$. Together with point (4) it follows that $(\mathbf{P}_k \mathbf{P}_l)^\top = -\mathbf{P}_k \mathbf{P}_l$. Similarly, from point (3) we have that $(\mathbf{P}_k \mathbf{P}_l)^\top = \mathbf{P}_l^\top \mathbf{P}_k^\top = (-\mathbf{P}_l)(-\mathbf{P}_k) = \mathbf{P}_l \mathbf{P}_k$. Therefore, the two matrices are anti-commutative: $\mathbf{P}_k \mathbf{P}_l = -\mathbf{P}_l \mathbf{P}_k$.

We remark that points (1), (3), (5) coincide with the defining algebraic properties of the algebra of complex numbers, quaternions, octonions, sedenions and their generalisations based on the Cayley-Dickson construction [59]. Therefore, the matrices in $\mathcal{P}$ must be a representation of one of these algebras. Firstly, such algebras exist only for $d$ that are powers of two. Secondly, matrix representations for these algebras exist only in dimensions two and four. This is because the algebra of octonions and their generalisations, unlike matrix multiplication, is non-associative. As a sanity check, note that the matrices $\mathbf{R}_1, \mathbf{R}_2$ from Lemma 27 are a well-known representation of the unit complex numbers.

We conclude this section by giving out the matrices for $d = 4$, which are the real matrix representations of the four unit quaternions:

$$
\mathbf{R}_1 = \begin{bmatrix} 1 & 0 & 0 & 0 \\ 0 & 1 & 0 & 0 \\ 0 & 0 & 1 & 0 \\ 0 & 0 & 0 & 1 \end{bmatrix}, \quad \mathbf{R}_2 = \begin{bmatrix} 0 & -1 & 0 & 0 \\ 1 & 0 & 0 & 0 \\ 0 & 0 & 0 & -1 \\ 0 & 0 & 1 & 0 \end{bmatrix}, \tag{19}
$$

$$
\mathbf{R}_3 = \begin{bmatrix} 0 & 0 & -1 & 0 \\ 0 & 0 & 0 & 1 \\ 1 & 0 & 0 & 0 \\ 0 & -1 & 0 & 0 \end{bmatrix}, \quad \mathbf{R}_4 = \begin{bmatrix} 0 & 0 & 0 & -1 \\ 0 & 0 & -1 & 0 \\ 0 & 1 & 0 & 0 \\ 1 & 0 & 0 & 0 \end{bmatrix}.
$$

It can be checked that these matrices respect the properties outlined above. Thus, in $d = 4$, we can select the transport maps from the set $\{\pm\mathbf{R}_1, \pm\mathbf{R}_2, \pm\mathbf{R}_3, \pm\mathbf{R}_4\}$ containing eight matrices, which also form a group. Therefore, following the same procedure as in Lemma 27, we can linearly separate up to eight classes. $\qquad\square$

**Proposition 28.** *Let $\mathcal{G}$ be the class of connected regular graphs with a finite number of classes. Then, $\mathcal{H}^2_{\mathrm{orth}}(G)$ has linear separation power over $\mathcal{G}$.*

*Proof.* Idea: Since the graph is regular, the harmonic features of the nodes will be uniformly scaled and thus positioned on a circle. The aim is to place different classes at different locations on the circle, which would make the classes linearly separable as shown in Figure 6b.

Let $G$ be a regular graph with $C$ classes and define $\theta = \frac{2\pi}{C}$. Denote by $\mathbf{R}_i$ the 2D rotation matrix:

$$
\mathbf{R}_i = \begin{bmatrix} \cos(i\theta) & -\sin(i\theta) \\ \sin(i\theta) & \cos(i\theta) \end{bmatrix} \tag{20}
$$

Then let $\mathcal{P} = \{\mathbf{R}_i \mid 0 \leq i \leq C - 1, \ i \in \mathbb{N}\}$ the set of rotation matrices with an angle multiple of $\theta$. Then we can define a bijection $g : \mathcal{C} \to \mathcal{P}$ and a sheaf $(G, \mathcal{F}) \in \mathcal{H}^2_{\mathrm{orth}}(G)$ as in the proof above. Checking the inner-products from Equation 18 between the harmonic features of the nodes, we can verify that the angle between any two classes is different from zero. By Lemma 26, the cross terms of the inner product vanish:

$$
\sum_k \alpha_k^2 \langle \mathbf{R}_i[k], \mathbf{R}_j[k] \rangle = \sum_k \alpha_k^2 \cos((i-j)\theta) = \cos((i-j)\theta)\|\mathbf{x}\|^2 \tag{21}
$$

Thus, the angle between classes $i, j$ is $(i - j)\theta$. $\qquad\square$

## C  Energy Flow Proofs

**Proposition 29.** *If $\mathcal{F}$ is an $O(d)$-bundle in $\mathcal{H}^d_{\mathrm{orth,sym}}$, then $\mathbf{x} \in \ker\Delta_{\mathcal{F}}$ if and only if $\mathbf{x}^k \in \ker\Delta_0$ for all $1 \leq k \leq d$.*

*Proof of Proposition 29.* Let $\mathbf{x} \in H^0(G, \mathcal{F})$. Then we have

$$
\begin{aligned}
0 = E_{\mathcal{F}}(\mathbf{x}) &= \frac{1}{2} \sum_{(v,u) \in E} ||\mathcal{F}_{v \trianglelefteq e} D_v^{-\frac{1}{2}} \mathbf{x}_v - \mathcal{F}_{u \trianglelefteq e} D_u^{-\frac{1}{2}} \mathbf{x}_u||^2 \\
&= \frac{1}{2} \sum_{(v,u) \in E} ||\mathcal{F}_e \left( D_v^{-\frac{1}{2}} \mathbf{x}_v - D_u^{-\frac{1}{2}} \mathbf{x}_u \right)||^2 \\
&= \frac{1}{2} \sum_{(v,u) \in E} ||d_v^{-\frac{1}{2}} \mathbf{x}_v - d_u^{-\frac{1}{2}} \mathbf{x}_u||^2.
\end{aligned}
$$

The last term vanishes if and only if $\mathbf{x}^k \in \ker \Delta_0$ for each $1 \leq k \leq d$. $\qquad\square$

**Proposition 17.** *For any connected graph $G$ and $\varepsilon > 0$, there exist a sheaf $(G, \mathcal{F}) \notin \mathcal{H}_{\text{sym}}^d$, $\mathbf{W}_1$ with $\|\mathbf{W}_1\|_2 < \varepsilon$ and feature vector $\mathbf{x}$ such that $E_{\mathcal{F}}((\mathbf{I} \otimes \mathbf{W}_1)\mathbf{x}) > E_{\mathcal{F}}(\mathbf{x})$.*

*Proof.* Let $\mathcal{F}$ be an $O(d)$-bundle over $G$ and $\varepsilon > 0$. Assume that $\mathcal{F}_{v \trianglelefteq e} = \mathcal{F}_{u \trianglelefteq e}$ for each $(u,v) \neq (u_0, v_0)$ and that $\mathcal{F}_{v_0 \trianglelefteq e}^{\top} \mathcal{F}_{u_0 \trianglelefteq e} - \mathbf{I} := \mathbf{B} \neq 0$ with $\dim(\ker(\mathbf{B})) > 0$. Then there exist a linear map $\mathbf{W} \in \mathbb{R}^{d \times d}$ with $\|\mathbf{W}\|_2 = \varepsilon$ and $\mathbf{x} \in H^0(G, \mathcal{F})$ such that $E_{\mathcal{F}}((\mathbf{I} \otimes \mathbf{W})\mathbf{x}) > 0$. We sketch the proof. Let $\mathbf{g} \in \ker(\mathbf{B})$. Define then $\mathbf{x} \in C^0(G, \mathcal{F})$ by

$$
\mathbf{x}_v = \sqrt{d_v} \mathbf{g}.
$$

Then $\mathbf{x} \in H^0(G, \mathcal{F})$. If we now take $\mathbf{W} = \varepsilon \mathbf{P}_{\ker \mathbf{B}^\perp}$ the rescaled orthogonal projection in the orthogonal complement of the kernel of $\mathbf{B}$ we verify the given claim. $\qquad\square$

We provide below a proof for the equality in Definition 14.

**Proposition 30.**

$$
\mathbf{x}^\top \Delta_{\mathcal{F}} \mathbf{x} = \frac{1}{2} \sum_{e := (v,u)} \|\mathcal{F}_{v \trianglelefteq e} D_v^{-1/2} \mathbf{x}_v - \mathcal{F}_{u \trianglelefteq e} D_u^{-1/2} \mathbf{x}_u\|_2^2
$$

*Proof.* We prove the result for the normalised sheaf Laplacian, and other versions can be obtained as particular cases.

$$
E(\mathbf{x}) = \mathbf{x}^\top \Delta_{\mathcal{F}} \mathbf{x} = \sum_v \mathbf{x}_v^\top \Delta_{vv} \mathbf{x}_v + \sum_{\substack{w \neq z \\ (w,z) \in E}} \mathbf{x}_w^\top \Delta_{wz} \mathbf{x}_z \tag{22}
$$

$$
= \sum_{v \trianglelefteq e} \mathbf{x}_v^\top D_v^{-1/2} \mathcal{F}_{v \trianglelefteq e}^\top \mathcal{F}_{v \trianglelefteq e} D_v^{-1/2} \mathbf{x}_v + \sum_{\substack{w < z \\ (w,z) \in E}} \mathbf{x}_w^\top \Delta_{wz} \mathbf{x}_z + \mathbf{x}_z^\top \Delta_{zw} \mathbf{x}_w \tag{23}
$$

$$
= \frac{1}{2} \sum_{v,w \trianglelefteq e} \left( \mathbf{x}_v^\top D_v^{-1/2} \mathcal{F}_{v \trianglelefteq e}^\top \mathcal{F}_{v \trianglelefteq e} D_v^{-1/2} \mathbf{x}_v + \mathbf{x}_w^\top D_w^{-1/2} \mathcal{F}_{w \trianglelefteq e}^\top \mathcal{F}_{w \trianglelefteq e} D_w^{-1/2} \mathbf{x}_w \right. \tag{24}
$$

$$
\left. + \mathbf{x}_v^\top D_v^{-1/2} \mathcal{F}_{v \trianglelefteq e}^\top \mathcal{F}_{w \trianglelefteq e} D_w^{-1/2} \mathbf{x}_w + \mathbf{x}_w^\top D_w^{-1/2} \mathcal{F}_{w \trianglelefteq e}^\top \mathcal{F}_{v \trianglelefteq e} D_v^{-1/2} \mathbf{x}_v \right) \tag{25}
$$

$$
= \frac{1}{2} \sum_{v,w \trianglelefteq e} \mathbf{x}_v^\top D_v^{-1/2} \mathcal{F}_{v \trianglelefteq e}^\top \left( \mathcal{F}_{v \trianglelefteq e} D_v^{-1/2} \mathbf{x}_v - \mathcal{F}_{w \trianglelefteq e} D_w^{-1/2} \mathbf{x}_w \right) \tag{26}
$$

$$
- \mathbf{x}_w^\top D_w^{-1/2} \mathcal{F}_{w \trianglelefteq e}^\top \left( \mathcal{F}_{v \trianglelefteq e} D_v^{-1/2} \mathbf{x}_v - \mathcal{F}_{w \trianglelefteq e} D_w^{-1/2} \mathbf{x}_w \right) \tag{27}
$$

$$
= \frac{1}{2} \sum_{v,w \trianglelefteq e} \left( \mathbf{x}_v^\top D_v^{-1/2} \mathcal{F}_{v \trianglelefteq e}^\top - \mathbf{x}_w^\top D_w^{-1/2} \mathcal{F}_{w \trianglelefteq e}^\top \right) \left( \mathcal{F}_{v \trianglelefteq e} D_v^{-1/2} \mathbf{x}_v - \mathcal{F}_{w \trianglelefteq e} D_w^{-1/2} \mathbf{x}_w \right) \tag{28}
$$

Note that $D_v$ is symmetric for any node $v$ and so is any $D_v^{-1/2}$. Therefore, the two vectors in the parenthesis are the transpose of each other and the result is their inner product. Thus, we have:

$$
E_{\mathcal{F}}(\mathbf{x}) = \frac{1}{2} \sum_{v,w \trianglelefteq e} \|\mathcal{F}_{v \trianglelefteq e} D_v^{-1/2} \mathbf{x}_v - \mathcal{F}_{w \trianglelefteq e} D_w^{-1/2} \mathbf{x}_w\|_2^2 \tag{29}
$$

The result follows identically for other types of Laplacian. For the augmented normalized Laplacian, one should simply replace $D$ with $\tilde{D} = D + I$ and for the non-normalised Laplacian, one should simply remove $D$ from the equation. $\qquad\square$

**Theorem 16.** *If $(\mathcal{F}, G) \in \mathcal{H}^d_{\mathrm{orth,sym}}$ and $\sigma = $ (Leaky)ReLU, $E_{\mathcal{F}}(\mathbf{Y}) \leq \lambda_* \|\mathbf{W}_1\|_2^2 \|\mathbf{W}_2^\top\|_2^2 E_{\mathcal{F}}(\mathbf{X})$.*

*Proof.* We first prove a couple of Lemmas before proving the Theorem. The proof structure follows that of Cai and Wang [13], which in turn generalises that of Oono and Suzuki [51]. The latter proof technique is not directly applicable to our setting because it makes some strong assumptions about the harmonic space of the Laplacian (i.e. that the eigenvectors of the harmonic space have positive entries).

$\lambda_* = \max\left((\lambda_{\min} - 1)^2, (\lambda_{\max} - 1)^2\right)$, where $\lambda_{\min}, \lambda_{\max}$ are the smallest and largest non-zero eigenvalues of $\Delta_{\mathcal{F}}$.

**Lemma 31.** *For $\mathbf{P} = \mathbf{I} - \Delta_{\mathcal{F}}$, $E_{\mathcal{F}}(\mathbf{Px}) < \lambda_* E_{\mathcal{F}}(\mathbf{x})$.*

*Proof.* We can write $\mathbf{x} = \sum_i c_i \mathbf{h}^i$ as a sum of the eigenvectors $\{\mathbf{h}^i\}$ of $\Delta_{\mathcal{F}}$. Then $\mathbf{x}^\top \Delta_{\mathcal{F}} \mathbf{x} = \sum_i c_i^2 \lambda_i$, where $\{\lambda_i\}$ are the eigenvalues of $\Delta_{\mathcal{F}}$.

$$E_{\mathcal{F}}(\mathbf{Px}) = \mathbf{x}^\top \mathbf{P}^\top \Delta_{\mathcal{F}} \mathbf{Px} = \mathbf{x}^\top \mathbf{P} \Delta_{\mathcal{F}} \mathbf{Px} = \sum_i c_i^2 \lambda_i (1 - \lambda_i)^2 \leq \lambda_* \sum_i c_i^2 \lambda_i = \lambda_* E_{\mathcal{F}}(\mathbf{x}) \quad (30)$$

The inequality follows from the fact that the eigenvectors of the normalised sheaf Laplacian are in the range $[0, 2]$ [34, Proposition 5.5]. We note that the original proof of Cai and Wang [13] bounds the expression by $(1 - \lambda_{\min})^2$ instead of $\lambda_*$, which appears to be an error. $\qquad\square$

**Lemma 32.** $E_{\mathcal{F}}(\mathbf{XW}) \leq \|\mathbf{W}^\top\|_2^2 E_{\mathcal{F}}(\mathbf{X})$

*Proof.* Following the proof of Cai and Wang [13] we have:

$$E_{\mathcal{F}}(\mathbf{XW}) = \mathrm{Tr}(\mathbf{W}^\top \mathbf{X}^\top \Delta_{\mathcal{F}} \mathbf{XW}) \tag{31}$$
$$= \mathrm{Tr}(\mathbf{X}^\top \Delta_{\mathcal{F}} \mathbf{XWW}^\top) \qquad \text{trace cyclic property} \tag{32}$$
$$\leq \mathrm{Tr}(\mathbf{X}^\top \Delta_{\mathcal{F}} \mathbf{X}) \|\mathbf{WW}^\top\|_2 \qquad \text{see Lemma 3.1 in Lee [40]} \tag{33}$$
$$= \mathrm{Tr}(\mathbf{X}^\top \Delta_{\mathcal{F}} \mathbf{X}) \|\mathbf{W}^\top\|_2^2 \tag{34}$$

$\qquad\square$

**Lemma 33.** *For conditions as in Theorem 16, $E_{\mathcal{F}}\left((\mathbf{I}_n \otimes \mathbf{W})\mathbf{x}\right) \leq \|\mathbf{W}\|_2^2 E_{\mathcal{F}}(\mathbf{x})$.*

*Proof.* First, we note that for orthogonal matrices, $D_v = \mathbf{I} \sum_{v \trianglelefteq e} \alpha_e^2 = \mathbf{I} d_v$ [34, Lemma 4.4]

$$E_{\mathcal{F}}\left((\mathbf{I} \otimes \mathbf{W})\mathbf{x}\right) = \frac{1}{2} \sum_{v, w \trianglelefteq e} \|\mathcal{F}_{v \trianglelefteq e} D_v^{-1/2} \mathbf{W} f_v - \mathcal{F}_{w \trianglelefteq e} D_w^{-1/2} \mathbf{W} \mathbf{x}_w\|_2^2 \tag{35}$$

$$= \frac{1}{2} \sum_{v, w \trianglelefteq e} \|\mathcal{F}_e \mathbf{W}\left(d_v^{-1/2} \mathbf{x}_v - d_w^{-1/2} \mathbf{x}_w\right)\|_2^2 \tag{36}$$

$$= \frac{1}{2} \sum_{v, w \trianglelefteq e} \|\mathbf{W}\left(d_v^{-1/2} \mathbf{x}_v - d_w^{-1/2} \mathbf{x}_w\right)\|_2^2 \qquad \mathcal{F}_e \text{ is orthogonal} \tag{37}$$

$$\leq \frac{1}{2} \sum_{v, w \trianglelefteq e} \|\mathbf{W}\|_2^2 \|d_v^{-1/2} \mathbf{x}_v - d_w^{-1/2} \mathbf{x}_w\|_2^2 \tag{38}$$

$$= \frac{1}{2} \sum_{v, w \trianglelefteq e} \|\mathbf{W}\|_2^2 \|\mathcal{F}_e\left(d_v^{-1/2} \mathbf{x}_v - d_w^{-1/2} \mathbf{x}_w\right)\|_2^2 \qquad \mathcal{F}_e \text{ is orthogonal} \tag{39}$$

$$= \frac{1}{2} \|\mathbf{W}\|_2^2 \sum_{v, w \trianglelefteq e} \|\mathcal{F}_e\left(D_v^{-1/2} \mathbf{x}_v - D_w^{-1/2} \mathbf{x}_w\right)\|_2^2 \tag{40}$$

$$= \|\mathbf{W}\|_2^2 E_{\mathcal{F}}(\mathbf{x}) \tag{41}$$

The proof can also be extended easily extended to vector bundles over weighted graphs (i.e. allowing weighted edges as in Hansen and Ghrist [34]). For the non-normalised Laplacian, the assumption that $\mathcal{F}_e$ is orthogonal can be relaxed to being non-singular and then the upper bound will also depend on the maximum conditioning number over all $\mathcal{F}_e$. $\qquad\square$

**Lemma 34.** *For conditions as in Theorem 16, $E_{\mathcal{F}}\big(\sigma(\mathbf{x})\big) \leq E_{\mathcal{F}}(\mathbf{x})$.*

*Proof.*

$$E\big(\sigma(\mathbf{x})\big) = \frac{1}{2}\sum_{v,w\trianglelefteq e}\|\mathcal{F}_{v\trianglelefteq e}D_v^{-1/2}\sigma(\mathbf{x}_v) - \mathcal{F}_{w\trianglelefteq e}D_w^{-1/2}\sigma(\mathbf{x}_w)\|_2^2 \tag{42}$$

$$= \frac{1}{2}\sum_{v,w\trianglelefteq e}\|\mathcal{F}_e\big(d_v^{-1/2}\sigma(\mathbf{x}_v) - d_w^{-1/2}\sigma(\mathbf{x}_w)\big)\|_2^2 \tag{43}$$

$$= \frac{1}{2}\sum_{v,w\trianglelefteq e}\|d_v^{-1/2}\sigma(\mathbf{x}_v) - d_w^{-1/2}\sigma(\mathbf{x}_w)\|_2^2 \qquad\text{orthogonality of }\mathcal{F}_e \tag{44}$$

$$= \frac{1}{2}\sum_{v,w\trianglelefteq e}\|\sigma\Big(\frac{\mathbf{x}_v}{\sqrt{d_v}}\Big) - \sigma\Big(\frac{\mathbf{x}_w}{\sqrt{d_w}}\Big)\|_2^2 \qquad c\mathrm{ReLU}(x) = \mathrm{ReLU}(cx), c > 0 \tag{45}$$

$$\leq \frac{1}{2}\sum_{v,w\trianglelefteq e}\|\frac{\mathbf{x}_v}{\sqrt{d_v}} - \frac{\mathbf{x}_w}{\sqrt{d_w}}\|_2^2 \qquad\text{Lipschitz continuity of ReLU} \tag{46}$$

$$= \frac{1}{2}\sum_{v,w\trianglelefteq e}\|\mathcal{F}_e\big(d_v^{-1/2}\mathbf{x}_v - d_w^{-1/2}\mathbf{x}_w\big)\|_2^2 \qquad\text{orthogonality of }\mathcal{F}_e \tag{47}$$

$$= E_{\mathcal{F}}(\mathbf{x}) \tag{48}$$

$\qquad\square$

Combining these three lemmas for an entire diffusion layer proves the Theorem. $\qquad\square$

**Theorem 15.** *For $(\mathcal{F}, G) \in \mathcal{H}_+^1$ and $\sigma$ being (Leaky)ReLU, $E_{\mathcal{F}}(\mathbf{Y}) \leq \lambda_*\|\mathbf{W}_1\|_2^2\|\mathbf{W}_2^\top\|_2^2 E_{\mathcal{F}}(\mathbf{X})$.*

*Proof.* If $d = 1$, then Lemma 33 becomes superfluous as $\mathbf{W}_1$ becomes a scalar that can be absorbed into the right-weights. It remains to verify that a version of Lemma 34 holds in this case.

**Lemma 35.** *For conditions as in Theorem 15, $E_{\mathcal{F}}\big(\sigma(\mathbf{x})\big) \leq E_{\mathcal{F}}(\mathbf{x})$.*

*Proof.*

$$E\big(\sigma(\mathbf{x})\big) = \frac{1}{2}\sum_{v,w\trianglelefteq e}\|\mathcal{F}_{v\trianglelefteq e}D_v^{-1/2}\sigma(x_v) - \mathcal{F}_{w\trianglelefteq e}D_w^{-1/2}\sigma(x_w)\|_2^2 \tag{49}$$

$$= \frac{1}{2}\sum_{v,w\trianglelefteq e}\||\mathcal{F}_{v\trianglelefteq e}|D_v^{-1/2}\sigma(x_v) - |\mathcal{F}_{w\trianglelefteq e}|D_w^{-1/2}\sigma(x_w)\|_2^2 \quad \mathcal{F}_{v\trianglelefteq e}\mathcal{F}_{w\trianglelefteq e} > 0 \tag{50}$$

$$= \frac{1}{2}\sum_{v,w\trianglelefteq e}\|\sigma\Big(\frac{|\mathcal{F}_{v\trianglelefteq e}|x_v}{\sqrt{d_v}}\Big) - \sigma\Big(\frac{|\mathcal{F}_{w\trianglelefteq e}|x_w}{\sqrt{d_w}}\Big)\|_2^2 \qquad c\sigma(x) = \sigma(cx), c > 0 \tag{51}$$

$$\leq \frac{1}{2}\sum_{v,w\trianglelefteq e}\|\frac{|\mathcal{F}_{v\trianglelefteq e}|x_v}{\sqrt{d_v}} - \frac{|\mathcal{F}_{w\trianglelefteq e}|x_w}{\sqrt{d_w}}\|_2^2 \qquad\text{ReLU Lipschitz cont.} \tag{52}$$

$$= \frac{1}{2}\sum_{v,w\trianglelefteq e}\|\mathcal{F}_{v\trianglelefteq e}D_v^{-1/2}x_v - \mathcal{F}_{w\trianglelefteq e}D_w^{-1/2}x_w\|_2^2 \qquad \mathcal{F}_{v\trianglelefteq e}\mathcal{F}_{w\trianglelefteq e} > 0 \tag{53}$$

$$= E_{\mathcal{F}}(\mathbf{x}) \tag{54}$$

$\qquad\square$

We note that if $\mathcal{F}_{v\trianglelefteq e}\mathcal{F}_{w\trianglelefteq e} < 0$ (i.e. the relation is signed), then it is very easy to find counter-examples where ReLU does not work anymore. However, the result still holds in the deep linear case. $\qquad\square$

If the features of an SCN/GCN oversmoothing as in Theorem 15 converge to $\ker(\Delta_{\mathcal{F}})$, then the model will no longer be able to linearly separate the classes. This is shown by the following Corollaries.

**Corollary 36.** *Consider an SCN model $f$ with $k$ layers and a sheaf $(\mathcal{F}; G) \in \mathcal{H}^1_{\mathrm{sym}}$ over a bipartite graph $G$ as in Proposition 9. Then for any finite $k$, $\mathbf{Y} := f(\mathbf{X})$ is not linearly separable for any input with $E_{\mathcal{F}}(\mathbf{X}) = 0$.*

*Proof.* By Theorem 15, if $E_{\mathcal{F}}(\mathbf{X}) = 0$, then $E_{\mathcal{F}}(\mathbf{Y}) = 0$ and, therefore, $\mathbf{Y}^i \in \ker(\Delta_{\mathcal{F}})$ for any column $i$. The proof of Proposition 9 showed that the classes of such a bipartite graph cannot be linearly separated for any such feature matrix $\mathbf{Y}$. $\qquad\square$

**Corollary 37.** *Consider an SCN model $f$ with $k$ layers and a sheaf $(\mathcal{F}; G) \in \mathcal{H}^1_+$ over any graph $G$ with more than two classes as in Proposition 11. Then for any finite $k$, $\mathbf{Y} := f(\mathbf{X})$ is not linearly separable for any input with $E_{\mathcal{F}}(\mathbf{X}) = 0$.*

*Proof.* By Theorem 15, if $E_{\mathcal{F}}(\mathbf{X}) = 0$, then $E_{\mathcal{F}}(\mathbf{Y}) = 0$ and, therefore, $\mathbf{Y}^i \in \ker(\Delta_{\mathcal{F}})$ for any column $i$. The proof of Proposition 11 showed that the classes of such a graph cannot be linearly separated for any such feature matrix $\mathbf{Y}$. $\qquad\square$

## D  Sheaf Learning Proof

**Proposition 18.** *Let $G = (V, E)$ be a finite graph with features $\mathbf{X}$. Then, if $(\mathbf{x}_v, \mathbf{x}_u) \neq (\mathbf{x}_w, \mathbf{x}_z)$ for any $(v, u) \neq (w, z) \in E$ and $\Phi$ is an MLP with sufficient capacity, $\Phi$ can learn any sheaf $(\mathcal{F}; G)$.*

*Proof.* Assume that the node features are $k$-dimensional and, therefore, the graph feature matrix has shape $\mathbf{X} \in \mathbb{R}^{n \times k}$. Define the finite set $A := \{(\mathbf{x}_v, \mathbf{x}_u) : v \to u \in E\} \subset \mathbb{R}^{2k}$ containing the concatenated features of the nodes for all the oriented edges $v \to u$ of the graph. Then, because each $(\mathbf{x}_v, \mathbf{x}_u)$ is unique, for any dimension $d$, there exists a (well-defined) function $g : A \to \mathbb{R}^{d \times d}$ sending $(\mathbf{x}_v, \mathbf{x}_u) \mapsto \mathcal{F}_{v \trianglelefteq e=(v,u)}$. We now show that this function can be extended to a smooth function $f : \mathbb{R}^{2k} \to \mathbb{R}^{d \times d}$ and, therefore, it can be approximated by an MLP due to the Universal Approximation Theorem [36, 37].

Let $I$ be an index set for the elements of $A$. Then, because $A$ is finite, for any $\mathbf{a}_{i \in I}$, we can find a sufficiently small neighbourhood $U_i \subset \mathbb{R}^{2k}$ such that $\mathbf{a}_i \in U_i$ and $\mathbf{a}_j \notin U_i$ for $j \neq i \in I$. Furthermore, for each $i \in I$, we can find a (smooth) bump function $\psi_i : \mathbb{R}^{2k} \to \mathbb{R}$ such that $\psi_i(\mathbf{a}_i) = 1$ and $\psi_i(\mathbf{a}) = 0$ if $\mathbf{a} \notin U_i$. Then, the function $f(\mathbf{a}) := \sum_{i \in I} g(\mathbf{a}_i)\psi_i(\mathbf{a})$ is smooth and $f|_A = g$. $\qquad\square$

## E  Additional model details and hyperparameters

**Hybrid transport maps.**  Consider the transport maps $-\mathcal{F}_{v \trianglelefteq e}^\top \mathcal{F}_{u \trianglelefteq e}$ appearing in the off-diagonal entires of the sheaf Laplacian $L_{\mathcal{F}}$. When learning a sheaf Laplacian, there exists the risk that the features are not sufficiently good in the early layers (or in general) and, therefore, it might be useful to consider a hybrid transport map of the form $-\mathcal{F}_{v \trianglelefteq e}^\top \mathcal{F}_{u \trianglelefteq e} \bigoplus \mathbf{F}$, where $\bigoplus$ is the direct sum of two matrices and $\mathbf{F}$ represents a fixed (non-learnable map). In particular, we consider maps of the form $-\mathcal{F}_{v \trianglelefteq e}^\top \mathcal{F}_{u \trianglelefteq e} \bigoplus \mathbf{I}_1 \bigoplus -\mathbf{I}_1$ which essentially appends a diagonal matrix with 1 and $-1$ on the diagonal to the learned matrix. From a signal processing perspective, these correspond to a low-pass and a high-pass filter that could produce generally useful features. We treat the addition of these fixed parts as an additional hyper-parameter.

**Adjusting the activation magnitudes.** We note that in practice we find it useful to learn an additional parameter $\varepsilon \in [-1, 1]^d$ (i.e. a vector of size $d$) in the discrete version of the models:

$$\mathbf{X}_{t+1} = (1 + \varepsilon)\mathbf{X}_t - \sigma\Big(\Delta_{\mathcal{F}(t)}(\mathbf{I} \otimes \mathbf{W}_1^t)\mathbf{X}_t\mathbf{W}_2^t\Big). \tag{55}$$

This allows the model to adjust the relative magnitude of the features in each stalk dimension. This is used across all of our experiments in the discrete models.

**Augmented normalised sheaf Laplacian.** Similarly to GCN which normalises the Laplacian by the augmented degrees (i.e. $(\mathbf{D} + \mathbf{I}_n)^{-1/2}$, where $\mathbf{D}$ is the usual diagonal matrix of node degrees), we similarly use $(D + \mathbf{I}_{nd})^{-1/2}$ for normalisation to obtain greater numerical stability. This is particularly helpful when learning general sheaves as it increases the numerical stability of SVD.

Table 2: Hyper-parameter ranges for the discrete and continous models.

|  | **Discrete Models** | **Continous Models** |
|---|---|---|
| Hidden channels | $(8, 16, 32)$ (WebKB) and $(8, 16, 32, 64)$ (others) | $(8, 16, 32, 64)$ |
| Stalk dim $d$ | $1 - 5$ | $1 - 5$ |
| Layers | $2 - 8$ | N/A |
| Learning rate | $0.02$ (WebKB) and $0.01$ (others) | Log-uniform $[0.01, 0.1]$ |
| Activation | ELU | ELU |
| Weight decay (regular parameters) | Log-uniform $[-4.5, 11.0]$ | Log-uniform $[-6.9, 13.8]$ |
| Weight decay (sheaf parameters) | Log-uniform $[-4.5, 11.0]$ | Log-uniform $[-6.9, 13.8]$ |
| Input dropout | Uniform $[0, 0.9]$ | Uniform $[0, 0.9]$ |
| Layer dropout | Uniform $[0, 0.9]$ | N/A |
| Patience (epochs) | 100 (Wiki) and 200 (others) | 50 |
| Max training epochs | 1000 (Wiki) and 500 (others) | 50. |
| Integration time | N/A | Uniform $[1.0, 9.0]$. |
| Optimiser | Adam [38] | Adam |

**Hyperparameters and training procedure.** We train all models for a fixed maximum number of epochs and perform early stopping when the validation metric has not improved for a pre-specified number of patience epochs. We report the results at the epoch where the best validation metric was obtained for the model configuration with the best validation score among all models. We use the hyperparameter optimisation tools provided by Weights and Biases [5] for this procedure. The complete hyperparameter ranges we optimised over can be found in Table 2. All models were trained and fine-tuned on an Amazon AWS p2.xlarge machine containing 8 NVIDIA K80 GPUs and using a 2.3 GHz (base) and 2.7 GHz (turbo) Intel Xeon E5-2686 v4 Processor.

### E.1 Computational Complexity

We can split the computational complexity into the following computational steps:

1. **The linear transformation $\mathbf{X}' = (\mathbf{I} \otimes \mathbf{W}_1^t)\mathbf{X}_t\mathbf{W}_2^t$.** $\mathbf{W}_1$ is a $d \times d$ matrix and $\mathbf{W}_2$ is an $f \times f$ matrix. Therefore, the complexity is $\mathcal{O}\left(n(d^2 f + df^2)\right) = \mathcal{O}\left(n(cd + cf)\right) = \mathcal{O}(nc^2)$.

2. **Message Passing.** Since $\Delta_{\mathcal{F}}$ is a sparse matrix, the message passing is implemented as a sparse-dense matrix multiplication $\Delta_{\mathcal{F}}\mathbf{X}'$. When the restriction maps are diagonal, the complexity of this operation is $\mathcal{O}(mc)$, since the multiplication of each block matrix in $\Delta_{\mathcal{F}}$ and block vector in $\mathbf{X}'$ reduces to a an element-wise vector multiplication. When the restriction maps are non-diagonal, the complexity is $\mathcal{O}(mdc)$ because each matrix-vector multiplication is $\mathcal{O}(d^2)$ and we need to perform $f$ of them for each node and edge.

3. **Learning the Sheaf.** Assume we learn the restriction maps via $\Phi(\mathbf{x}_v, \mathbf{x}_u) = \sigma(\mathbf{V}[\text{vec}(\mathbf{X}_v)||\text{vec}(\mathbf{X}_u)])$, where $\text{vec}(\cdot)$ converts the $d \times f$ matrix into a $df$-sized vector. This operation has to be performed for each incident node-edge pair. Therefore, the complexity is $\mathcal{O}(md^2 f) = \mathcal{O}(mcd)$, when learning diagonal maps since $\mathbf{V}$ is a $d \times 2df$ matrix. When learning a non-diagonal matrix, the number of rows of $\mathbf{V}$ is $\mathcal{O}(d^2)$ and the complexity becomes $\mathcal{O}(md^3 f) = \mathcal{O}(mcd^2)$. Note, however, that in general, the complexity of learning the restriction maps can be significantly reduced to $\mathcal{O}(mc)$ (in the diagonal case) and $\mathcal{O}(m(c + d^2))$ (in the non-diagonal case) by, for instance, using an MLP with constant hidden-size.

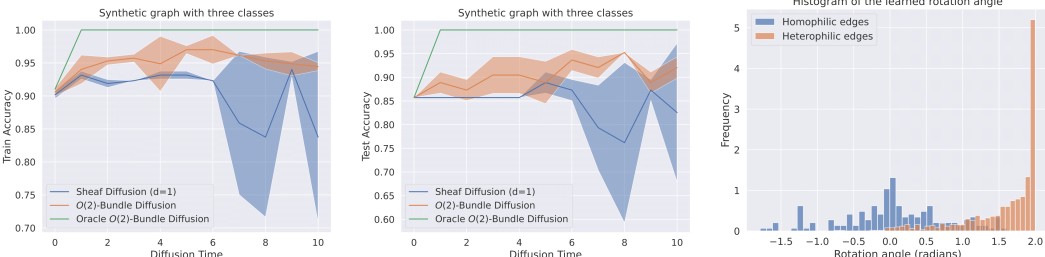

Figure 7: (*Left*) Train accuracy as a function of diffusion time. (*Middle*) Test accuracy as a function of diffusion time. (*Right*) Histogram of the learned rotation angle of the $2D$ transport maps. The performance of the bundle model is superior to that of the one-dimensional sheaf. The transport maps learned by the model are aligned with our expectation: the model learns to rotate more (i.e. to move away) the neighbours belonging to different classes than the neighbours belonging to the same class.

4. **Constructing the Laplacian.** To build the Laplacian, we need to perform the matrix-matrix multiplications involved in computing each of the blocks. The complexity of that is $\mathcal{O}(md)$ in the diagonal case and $\mathcal{O}(md^3)$ in the non-diagonal case. Computing the normalisation of the Laplacian is $\mathcal{O}(nd)$ in the diagonal case and $\mathcal{O}(nd^3)$ in the non-diagonal case.

Putting everything together, the final complexity is $\mathcal{O}(nc^2 + mcd)$ in the diagonal case and $\mathcal{O}\left(n(c^2 + d^3) + m(cd^2 + d^3)\right)$ in the non-diagonal case. When learning the sheaf via an MLP with constant hidden size, the complexity reduces to $\mathcal{O}(nc^2 + mc)$ (same as GCN) and $\mathcal{O}\left(n(c^2 + d^3) + m(c + d^3)\right)$, respectively.

For learning orthogonal matrices, we rely on the library Torch Householder [49] which provides support for fast transformations with large batch sizes.

## F   Additional Experiments

In this section, we provide a series of additional experiments and ablation studies.

**Two-dimensional synthetic experiment.**   In the main text we focused on a synthetic example involving sheaves with one-dimensional stalks. We now consider a graph with three classes and two-dimensional features, with edge homophily level $0.2$. We use $80\%$ of the nodes for training and $20\%$ for testing. First, we know that a discrete vector bundle with two-dimensional stalks that can solve the task in the limit exists from Theorem 13, while based on Proposition 11 no sheaf with one-dimensional stalks can perfectly solve the tasks.

Therefore, similarly to the synthetic experiment in the main text, we compare two similar models learning the sheaf from data: one using $1D$ stalks and another using $2D$ stalks. As we see from Figure 7, the discrete vector bundle model has better training and test-time performance than the one-dimensional counterpart. Nonetheless, none of the two models manages to match the perfect performance of the ideal sheaf on this more challenging dataset. From the final subfigure, we also see that the model learns to rotate more across the heterophilic edges in order to push away the nodes belonging to other classes. The prevalent angle of this rotation is 2 radians, which is just under $120° = 360°/C$, where $C = 3$ is the number of classes. Thus the model learns to position the three classes at approximately equal arc lengths from each other for maximum linear separability.

**Continous Models.**   To also understand how the continuous version of our models performs against other PDE-based GNNs we include a category of such SOTA models: CGNNs [70], GRAND [15], and BLEND [14]. Results are included in Table 3. Generally, continuous models do not perform as well as the discrete ones because they are constrained to use the same set of weights for the entire integration time and cannot use dropout. Therefore, the model capacity is difficult to increase without overfitting. Nonetheless, our continuous models generally outperform other state-of-the-art continuous models, which also share the same limitations. Finally, we note that the baselines were fine-tuned over the same hyper-parameter ranges as in Table 2

Table 3: Results on node classification datasets sorted by their homophily level. Top three models are coloured by **First**, **Second**, **Third**. Our models are marked **NSD**.

| | Texas | Wisconsin | Film | Squirrel | Chameleon | Cornell | Citeseer | Pubmed | Cora |
|---|---|---|---|---|---|---|---|---|---|
| Hom level | **0.11** | **0.21** | **0.22** | **0.22** | **0.23** | **0.30** | **0.74** | **0.80** | **0.81** |
| #Nodes | 183 | 251 | 7,600 | 5,201 | 2,277 | 183 | 3,327 | 18,717 | 2,708 |
| #Edges | 295 | 466 | 26,752 | 198,493 | 31,421 | 280 | 4,676 | 44,327 | 5,278 |
| #Classes | 5 | 5 | 5 | 5 | 5 | 5 | 7 | 3 | 6 |
| **Cont Diag-NSD** | $82.97_{\pm4.37}$ | $\mathbf{86.47}_{\pm2.55}$ | $\mathbf{36.85}_{\pm1.21}$ | $38.17_{\pm9.29}$ | $62.06_{\pm3.84}$ | $80.00_{\pm6.07}$ | $76.56_{\pm1.19}$ | $\mathbf{89.47}_{\pm0.42}$ | $86.88_{\pm1.21}$ |
| **Cont O(d)-NSD** | $82.43_{\pm5.95}$ | $84.50_{\pm4.34}$ | $36.39_{\pm1.37}$ | $40.40_{\pm2.01}$ | $63.18_{\pm1.69}$ | $72.16_{\pm10.40}$ | $75.19_{\pm1.67}$ | $89.12_{\pm0.30}$ | $86.70_{\pm1.24}$ |
| **Cont Gen-NSD** | $83.78_{\pm6.62}$ | $85.29_{\pm3.31}$ | $\mathbf{37.28}_{\pm0.74}$ | $52.57_{\pm2.76}$ | $\mathbf{66.40}_{\pm2.28}$ | $84.60_{\pm4.69}$ | $77.54_{\pm1.72}$ | $\mathbf{89.67}_{\pm0.40}$ | $\mathbf{87.45}_{\pm0.99}$ |
| BLEND | $\mathbf{83.24}_{\pm4.65}$ | $84.12_{\pm3.56}$ | $35.63_{\pm0.89}$ | $\mathbf{43.06}_{\pm1.39}$ | $60.11_{\pm2.09}$ | $\mathbf{85.95}_{\pm6.82}$ | $\mathbf{76.63}_{\pm1.60}$ | $89.24_{\pm0.42}$ | $\mathbf{88.09}_{\pm1.22}$ |
| GRAND | $75.68_{\pm7.25}$ | $79.41_{\pm3.64}$ | $35.62_{\pm1.01}$ | $40.05_{\pm1.50}$ | $54.67_{\pm2.54}$ | $82.16_{\pm7.09}$ | $76.46_{\pm1.77}$ | $89.02_{\pm0.51}$ | $87.36_{\pm0.96}$ |
| CGNN | $71.35_{\pm4.05}$ | $74.31_{\pm7.26}$ | $35.95_{\pm0.86}$ | $29.24_{\pm1.09}$ | $46.89_{\pm1.66}$ | $66.22_{\pm7.69}$ | $\mathbf{76.91}_{\pm1.81}$ | $87.70_{\pm0.49}$ | $87.10_{\pm1.35}$ |

**Positional encoding ablation.** Based on Proposition 18 we proceed to analyse the impact of increasing the expressive power of the model by making the nodes more distinguishable. For that, we equip our datasets with additional features consisting of graph Laplacian positional encodings as originally done in Dwivedi et al. [23]. In Table 4 we see that positional encodings do indeed improve the performance of the continuous models compared to the numbers reported in the main table. Therefore, we conclude that the interaction between the problem of sheaf learning and that of the expressivity of graph neural networks represents a promising avenue for future research.

Table 4: Ablation study for positional encodings. Positional encodings improve performance on some of our models.

| | Eigenvectors | Texas | Wisconsin | Cornell |
|---|---|---|---|---|
| Cont Diag-SD | 0 | $82.97 \pm 4.37$ | $\mathbf{86.47 \pm 2.55}$ | $80.00 \pm 6.07$ |
| | 2 | $3.51 \pm 5.05$ | $85.69 \pm 3.73$ | $81.62 \pm 8.00$ |
| | 8 | $\mathbf{85.41 \pm 5.82}$ | $86.28 \pm 3.40$ | $\mathbf{82.16 \pm 5.57}$ |
| | 16 | $82.70 \pm 3.86$ | $85.88 \pm 2.75$ | $81.08 \pm 7.25$ |
| Cont $O(d)$-SD | 0 | $82.43 \pm 5.95$ | $84.50 \pm 4.34$ | $72.16 \pm 10.40$ |
| | 2 | $84.05 \pm 5.85$ | $85.88 \pm 4.62$ | $83.51 \pm 9.70$ |
| | 8 | $\mathbf{84.87 \pm 4.71}$ | $\mathbf{86.86 \pm 3.83}$ | $\mathbf{84.05 \pm 5.85}$ |
| | 16 | $83.78 \pm 6.16$ | $85.88 \pm 2.88$ | $83.51 \pm 6.22$ |
| Cont Gen-SD | 0 | $\mathbf{83.78 \pm 6.62}$ | $85.29 \pm 3.31$ | $\mathbf{84.60 \pm 4.69}$ |
| | 2 | $83.24 \pm 4.32$ | $84.12 \pm 3.97$ | $81.08 \pm 7.35$ |
| | 8 | $82.70 \pm 5.70$ | $84.71 \pm 3.80$ | $83.24 \pm 6.82$ |
| | 16 | $82.16 \pm 6.19$ | $\mathbf{86.47 \pm 3.09}$ | $82.16 \pm 6.07$ |

**Visualising diffusion.** To develop a better intuition of the limiting behaviour of sheaf diffusion for node classification tasks we plot the diffusion process using an oracle discrete vector bundle for two graphs with $C = 3$ (Figure 9) and $C = 4$ (Figure 8) classes. The diffusion processes converge in the limit to a configuration where the classes are rotated at $\frac{2\pi}{C}$ from each other, just like in the cartoon diagrams of Figure 6. Note that in all cases, the classes are linearly separable in the limit.

We note that this approach generalises to any number of classes, but beyond $C = 4$ it is not guaranteed that they will be linearly separable in $2D$. However, they are still well separated. We include an example with $C = 10$ classes in Figure 10.

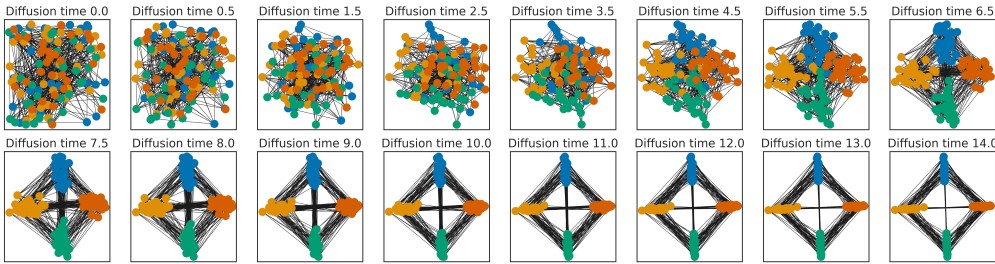

Figure 8: Sheaf diffusion process disentangling the $C = 4$ classes over time. The nodes are coloured by their class.

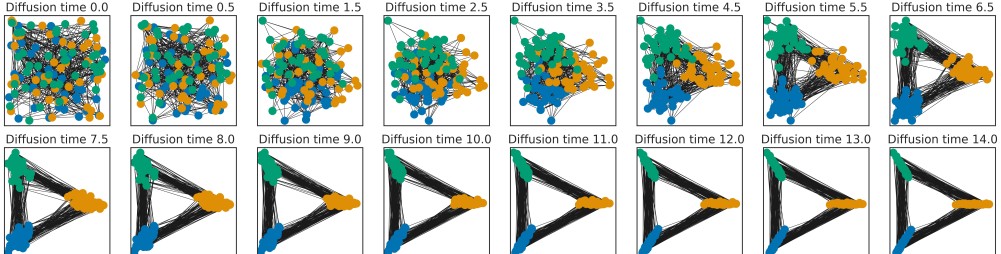

Figure 9: Sheaf diffusion process disentangling the $C = 3$ classes over time. The nodes are coloured by their class.

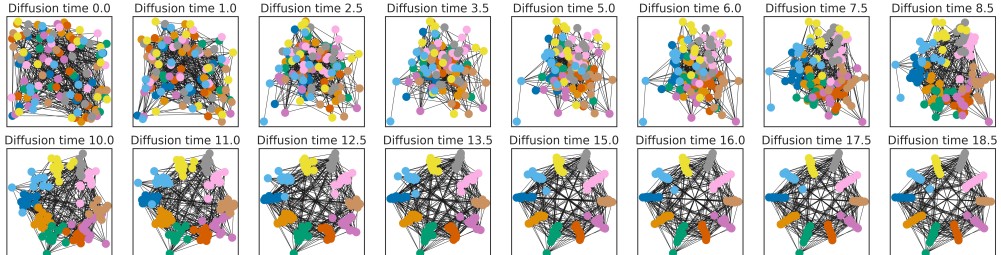

Figure 10: Sheaf diffusion process disentangling the $C = 10$ classes over time. The nodes are coloured by their class.