# OpenReview forum: "Neural Sheaf Diffusion: A Topological Perspective on Heterophily and Oversmoothing in GNNs"
_NeurIPS.cc/2022/Conference — NeurIPS 2022 Accept_

### Official Review · Reviewer_dwBG · 2022-07-11

**Rating:** 7
**Confidence:** 3
**Soundness:** 4 excellent
**Presentation:** 3 good
**Contribution:** 4 excellent

**Summary:**

This paper investigates the heterophily and oversmoothing problems in graph neural networks through the lens of cellular sheaf theory, and reveals that these problems can be avoided by building a proper sheaf model upon graph that is suitable for the task. It also proposes NSD model based on the theory, which is latter verified effective in heterophilic graphs.

**Questions:**

See above.

**Limitations:**

I don’t think this paper has any negative societal impact.

**Strengths And Weaknesses:**

In general, the idea of building a connection of sheaf theory and graph neural networks is somehow intriguing, and perhaps also fruitful in the sense that it provides a new perspective to understand heterophily and oversmoothing problems. The corresponding theoretical results about how a proper choice of the sheaf structure can help to solve these problems may greatly benefit our community. The mathematical analysis seem extensive and sound though I didn’t check the proof in the supplementary material. Experimental results also show that the proposed NSD model performs better especially in heterophilic graphs, which is in alignment with the theoretical findings.

I think the downside of this paper is the readability. Since it is very likely that most of the interested readers from the graph learning community are not familiar with sheaf theory (or differential geometry and algebraic topology), the readability of this paper would improve greatly to add more intuitive explanations on abstract theoretical results, e.g., regarding why change of sheaf structure can affect model’s separation power. In addition, it appears that the theoretical results in Section 3 about the properties of harmonic space are only of analysis interest and their connection with rest of the paper is somehow loose. I suggest the authors to move some of the details to appendix or explain more about their connections if they are indeed crucial.

I also have some concerns or questions regarding the implementation and experiments.
1. The computation complexity seems to be large, which scales with the stalk dimension as O(d^3). Though one can use batched GPU computations for training as the authors pointed out, the scalability problem may still remain when in comes to making predictions in the inference stage.
2. The paper seems to be missing a baseline, i.e., SheafNN [1], the predecessor of the proposed NSD. The authors also claimed that the proposed NSD is more powerful than SheafNN, so to what extent can NSD empirically outperform SheafNN?
3. The authors claimed in Section 4 that higher stalk dimension and more complex restriction maps indicate better separation power. But does better separation power necessarily translate into better ability to handle heterophily? If this is the case, how does the performance of NSD change with increasing d and restriction map complexity?
4. Table 4 shows the results of adding positional encodings, but what does it have to do with stalk dimension in the table captions? (p.s., ‘3.51 ± 5.05’ in the table seems to be wrong?)

[1] Sheaf neural networks. In NeurIPS 2020 Workshop.

---

> ### Author Response · Authors · 2022-07-31
> **Response to Reviewer dwBG**
>
> > **The computation complexity seems to be large, which scales with the stalk dimension as O(d^3). [...] the scalability problem may still remain when in comes to making predictions in the inference stage.**
>
> First, the complexity in the diagonal case scales with $O(d)$ and even assuming $d=1$, the proposed models still provide a wider class of Laplacians (and associated convolutions) than what existing models currently use with no increase in complexity. Furthermore, $d$ is typically small (i.e. 1-5) and one has the flexibility to make the restriction maps as sparse as they wish. For instance, the restriction maps could be made block diagonal or tridiagonal. Therefore, there is full flexibility to adjust the complexity anywhere between $O(d)$ and $O(d^3)$.
>
> Second, if at inference time the features & graph do not change, then it is possible to simply save / store the Laplacian(s) learned during training. Then, even in the general case, the complexity becomes $O(d^2)$ because the $O(d^3)$ matrix-matrix multiplications required in the Laplacian construction are no longer needed and one only has to do matrix-vector multiplications, which are $O(d^2)$.
>
> > **The paper seems to be missing a baseline, i.e., SheafNN [1], the predecessor of the proposed NSD. The authors also claimed that the proposed NSD is more powerful than SheafNN, so to what extent can NSD empirically outperform SheafNN?**
>
> As mentioned in the paper, SheafNN was proposed for settings where the sheaf structure is known. For real-world datasets this is not known, so the question is: what sheaf structure should we have used for the SheafNN? There is only one reasonable choice in the absence of any domain knowledge: the constant sheaf given by stalks isomorphic to $\mathbb{R}$ and restrictions equal to the identity. This choice of sheaf actually yields GCN, as we also mention in the paper, and we evaluate against GCN.
>
> > **I think the downside of this paper is the readability. Since it is very likely that most of the interested readers from the graph learning community are not familiar with sheaf theory, the readability of this paper would improve greatly to add more intuitive explanations on abstract theoretical results.**
>
> Readability can always be improved and we will do our best to implement the suggested changes in the additional space we will have for the final version.
>
> At the same time, we want to remark that this is an extremely difficult task in the hope that the reviewer will appreciate the huge efforts we have made in this direction already. Sheaf theory is a highly sophisticated topic (even for mathematicians) and relies on very abstract areas of mathematics such as category theory and homological algebra. As it is (we hope) clear from the paper, we have put a great deal of effort into getting rid of most of this advanced mathematical jargon and expressing everything in terms of linear algebra and opinion-dynamics-based intuitions, which should be accessible to an ML audience.
>
> Finally, to improve readability even further, we are currently writing a short introduction to sheaf theory for ML practitioners that we plan to add to the appendix in the hope that it will make the paper even friendlier.
>
> > **The authors claimed in Section 4 that higher stalk dimension and more complex restriction maps indicate better separation power. But does better separation power necessarily translate into better ability to handle heterophily? If this is the case, how does the performance of NSD change with increasing d and restriction map complexity?**
>
> The short answer is no. More complex restriction maps essentially imply a larger hypothesis space for our model and our theoretical results essentially explain the benefits of this larger hypothesis space and the limitations of comparatively smaller hypothesis classes. At the same time, as we know from standard statistical learning theory, a larger hypothesis space can also lead to overfitting and, therefore, it does not necessarily lead to better generalisation performance.
>
> The results in our experiments are consistent with this viewpoint. The main table shows the model with orthogonal restriction maps performs best overall since it finds the right middle ground between the very simple diagonal maps and the arbitrarily general restriction maps. Similarly, we notice a similar behaviour with the stalk dimension. The performance initially increases as $d$ is increased, but it starts to drop due to overfitting once $d$ becomes too large.
>
> > **Table 4 shows the results of adding positional encodings, but what does it have to do with stalk dimension in the table captions? (p.s., ‘3.51 ± 5.05’ in the table seems to be wrong?)**
>
> The caption of the table is an error on our side and we will fix it. Indeed, the table shows an ablation study about positional encodings. And yes, the number in the table should be “83.51 ± 5.05”. We will fix this as well. Thank you for pointing these out!

---

> > ### Comment · Reviewer_dwBG · 2022-08-08
> > **Thank you for your detailed rebuttal**
> >
> > I thank the authors for the detailed rebuttal, which has addressed most of my concerns. Therefore, I would keep my original score and recommend acceptance. : )

---

### Official Review · Reviewer_s77q · 2022-07-11

**Rating:** 9
**Confidence:** 4
**Soundness:** 4 excellent
**Presentation:** 4 excellent
**Contribution:** 4 excellent

**Summary:**

This paper uses cellular sheaf theory to study the oversmoothing behavior of GCNs and their limitations in heterophilic graphs. Additionally, it proposes a novel deep convolutional model, the SCN, and extension of Kipf and Welling's GCN where one simultaneously learns the sheaf and the convolution weights.

**Questions:**

- Per the theorem statement, Theorem 18 only holds for the Leaky ReLU and ReLU nonlinearities. Would other nonlinearities (e.g., the sigmoid or the tanh) significantly change its result?
- To improve the readability of the paper, a minor suggestion is to make the conclusions in lines 239-242 (that GCNs cannot linearly separate heterophilic bipartite graphs and graphs with more than two classes respectively) corollaries, or to highlight them somehow.

**Limitations:**

The main limitation of the paper is that its analysis only holds for the asymptotic case where $L\to\infty$ ($L$ is the number of layers). In well-connected graphs, using many message-passing layers leads to redundancy, so it is not hard to see how a deep architecture might exhibit oversmoothing. The authors should discuss this limitation more thoroughly (both in Section 8 and in the introduction), especially considering the fact that, in practice, many of the GCN architectures that exhibit good performance are not very deep.

**Strengths And Weaknesses:**

Strengths:

- The paper is clear, well-written, and easy to understand (especially given the complexity of the topic).

- The technical contributions are timely and significant: Proposition 10 and Theorem 18 provide a possible explanation for why GCNs do poorly in node classification problems on heterophilic graphs, and Proposition 12 and Theorem 18 for why they do poorly in problems with more than two classes. In contrast, Proposition 13 and Theorem 16 show that general SCNs do not have these limitations because neural sheaf diffusion can achieve linear class separation, and that this separation can be achieved exponentially fast. At the same time, Proposition 17 shows that in SCNs the Dirichlet energy is not constrained to decrease (unlike in GCNs).

Weaknesses:

- The paper (specifically, Sections 4 and 5) lacks a more precise link between the specific theoretical results (propositions, theorems) and the specific problems (limitations associated with heterophily, oversmoothing) that they help explain. In particular, the link between the Dirichlet energy and oversmoothing can be made clearer. Another suggestion is to indicate, in the introduction or in the gray inserts, which theorems/propositions explain the heterophily and oversmoothing problems respectively.

- The numerical experiments are convincing but limited.

     - Synthetic experiments: in the main body of the paper, it would be more interesting to see results for $d>1$. The benefits of
       asymmetric Laplacians and negative weights have been previously observed in the literature. Moreover, considering $d>1$ would
       provide a better illustration of the generality of the model, as SCNs with $d>1$ cannot be reduced to GCNs. One suggestion is to
       compare the ability of SCNs and GCNs to perform node classification on graphs with more than two classes.

     - Real-world experiments: the gains in performance (w.r.t. to other GNN models) in Table 1 are not very significant, but the results are
       competitive and the main contribution of the paper are its theoretical results.

---

> ### Author Response · Authors · 2022-07-31
> **Response to Reviewer s77q**
>
> > **The main limitation of the paper is that its analysis only holds for the asymptotic case where $L \rightarrow \infty$ ($L$ is the number of layers). In well-connected graphs, using many message-passing layers leads to redundancy, so it is not hard to see how a deep architecture might exhibit oversmoothing.**
>
> We want to remark that sheaf diffusion and other discrete processes discussed in the paper converge exponentially fast to the limiting behaviour. Therefore, everything we discuss is applicable to models with a reasonable number of layers (e.g. 5-10). We will emphasise this in the final version.
>
> > **Per the theorem statement, Theorem 18 only holds for the Leaky ReLU and ReLU nonlinearities. Would other nonlinearities (e.g., the sigmoid or the tanh) significantly change its result?**
>
> The result relies on the fact that choosing the nonlinearity $\sigma$ as ReLU or Leaky ReLU satisfies $c\sigma(x) = \sigma(cx)$ for $c >0$ and it is easy to see this does not hold for other often used nonlinearities.
>
> > **To improve the readability of the paper, a minor suggestion is to make the conclusions in lines 239-242 (that GCNs cannot linearly separate heterophilic bipartite graphs and graphs with more than two classes respectively) corollaries, or to highlight them somehow.**
>
> We will use the additional space available for the final version to do this.
>
> > **Synthetic experiments: in the main body of the paper, it would be more interesting to see results for d > 1 [...] One suggestion is to compare the ability of SCNs and GCNs to perform node classification on graphs with more than two classes.**
>
> We include a multi-class synthetic experiment with $d=2$ in Appendix F (see Figure 6). Unfortunately, due to space limitations, we did not manage to include this in the main text but we will better signpost this experiment in the main text.
>
> > **The link between the Dirichlet energy and oversmoothing can be made clearer. Another suggestion is to indicate, in the introduction or in the gray inserts, which theorems/propositions explain the heterophily and oversmoothing problems respectively.**
>
> We will address these aspects in the final version. Thank you!

---

> > ### Comment · Reviewer_s77q · 2022-08-03
> > **All of my comments have been addressed**
> >
> > Thank you for addressing my comments. I stand by my original evaluation of the paper.

---

### Official Review · Reviewer_2nUo · 2022-07-12

**Rating:** 6
**Confidence:** 4
**Soundness:** 3 good
**Presentation:** 3 good
**Contribution:** 3 good

**Summary:**

In this paper, the authors define a study a notion of graph neural networks that involves a notion of "sheafs" on graphs, which are vector spaces and linear operators attached to the nodes and edges of the graph. A givencellular sheaf thus defines a particular notion of message passing, where messages are transformed by the linear maps during propagation. This defines a generalized notion of Laplacian, as a block matrice involving the linear operators. The authors perform a basic harmonic study of this Laplacian, whose dominant eigenvector determines the infinite-time limit of the corresponding diffusion process. They study different classes of restricted sheafs, and their power to separate data by this diffusion process. Finally, a sheaf learning procedure is presented, similar to a generalized attention process.

**Questions:**

- in practice, there is no guarantee to learn effectively the right sheaf that will separate the data, is your asymptotic analysis robust to perturbed sheaf?
- if you learn only a single for all layers to simulate the diffusion process, do you learn what is expected in your analysis? Eg on simple two-communities graphs?

**Limitations:**

Discussion is appropriate.

**Strengths And Weaknesses:**

Strength:
- an original notion, grounded in algebraic geometry and topology, with an original instanciation on graphs
- a quite complete investigation

Weaknesses:
- the paper could be more explicit regarding the relationship with the proposed model and existing ones. In particular, when the sheaf must be learned, unless I am wrong the proposed model falls under the classical paradigm of message-passing (but with a generalized notion of "attention" that compute matrices between nodes). So, all the Weisfeiler Lehman analysis still applies (for instance), and statements like the end of Sec 5 "SCNs are more expressive than GCNs" are a bit misleading.
- Regarding this, the proposed architecture probably brings some implicit bias to the training, but it is difficult to which extent this really is more "powerful" than classical MPNNs (it seems SCNs could be, for instance, simulated with enough layers and MLPs). Moreover, in practice a different "sheaf" is learned at each layer, so it is not clear how the limit diffusion analysis is really observed in practice.

---

> ### Author Response · Authors · 2022-07-31
> **Response to Reviewer 2nUo**
>
> > **In practice, there is no guarantee to learn effectively the right sheaf that will separate the data, is your asymptotic analysis robust to perturbed sheaf?**
>
> This is the purpose of our analysis in Section 5. We showed that sheaf convolutions are not generally constrained to asymptotically end up in the kernel of the Laplacian (but can easily do so; and this is indeed desirable if the right sheaf is found). Therefore, if the model learns only a sheaf that is approximately right or, in the worst case, a sheaf that is completely wrong, it has (unlike GCNs) some flexibility to adjust their asymptotic behaviour via the weights $W_1$ and $W_2$.
>
> > **If you learn only a single [sheaf] for all layers to simulate the diffusion process, do you learn what is expected in your analysis? Eg on simple two-communities graphs?**
>
> This is what we did in the synthetic experiment in Figure 4. There we used a vanilla diffusion process, i.e. no weights and nonlinearities with a learned Laplacian that remains the same at all times, to classify the partitions of a bipartite graph. When inspecting the transport maps learned by the model in the third subfigure, we see that all of them are negative, as we would expect based on Proposition 3.1.
>
> We also did a similar higher dimensional experiment in Appendix F (see Figure 6) where we used a graph with high heterophily and three communities. On this graph, we learn a vector bundle. When inspecting the learned bundle, we see that the model generally learned (as expected) to rotate by a large angle when transporting the features across heterophilic edges and used an angle closer to zero for homophilic edges.
>
> > **The paper could be more explicit regarding the relationship with the proposed model and existing ones. In particular, when the sheaf must be learned, unless I am wrong the proposed model falls under the classical paradigm of message-passing (but with a generalized notion of "attention" that compute matrices between nodes). So, all the Weisfeiler Lehman analysis still applies (for instance), and statements like the end of Sec 5 "SCNs are more expressive than GCNs" are a bit misleading.**
>
> When we say "SCNs are more expressive than GCNs" we mean in the precise sense of our analysis (i.e. it is easier for SCNs to avoid oversmoothing) and not in the WL sense. The WL test is only one way to analyse the expressive power of GNNs and it has many limitations such as the fact that it relies on a countable feature alphabet and it is based on the problem of graph isomorphism, which is arguably not particularly relevant in a node classification setting where we work with only one graph. In contrast, our expressivity analysis is based on dynamical systems and the ability of certain dynamics to linearly separate the nodes. This new way of assessing the expressive power of GNNs is one of the main contributions of the paper.
>
> We would also like to clarify that our analysis does not concern all MPNNs, but it is restricted to various classes of sheaf diffusion processes and their associated convolutions, among which GCNs are a particular case. However, we agree that we could expand more on the relation between sheaf-based models and other MPNNs such as attention-based models and we will make use of the additional space for the final version to do so.
>
> > **Regarding this, the proposed architecture probably brings some implicit bias to the training, but it is difficult to which extent this really is more "powerful" than classical MPNNs (it seems SCNs could be, for instance, simulated with enough layers and MLPs).**
>
> While SCNs fall in the class of MPNNs, not all MPNNs are equivalent in terms of expressive power and particularly so on tasks like node classification where the node features are essential. In this regard, we believe we have provided theoretical guarantees for why SCNs might generally have more expressive power than GCNs in terms of linear separation capabilities. In the general WL landscape though, SCNs are MPNNs and we never claimed they are more powerful than (all) MPNNs.
>
> > **In practice a different "sheaf" is learned at each layer, so it is not clear how the limit diffusion analysis is really observed in practice.**
>
> The fact that we use something more general in practice is not an impediment. The model could learn in principle the same sheaf at each layer or could simply be restricted to using the same sheaf after a certain number of layers.
>
> The gist of our results is somewhat similar to the Universal Approximation Theorem, which tells us there is an MLP that can approximate any continuous function but finding it is a completely different problem which requires taking many things into account such as the optimisation procedure, initialisation and so on. Similarly, we only prove the existence of certain sheaf models that can solve the task, but finding them is another matter.

---

> > ### Comment · Reviewer_2nUo · 2022-08-08
> > **Response to rebuttal**
> >
> > I thank the authors for their rebuttal.
> > I am quite satisfied with the answers, but I will keep my score which I think is a fair assessment. I am confident that the paper will be accepted given the other good scores.

---

### Official Review · Reviewer_QFM1 · 2022-07-12

**Rating:** 6
**Confidence:** 3
**Soundness:** 3 good
**Presentation:** 3 good
**Contribution:** 3 good

**Summary:**

The paper proposes a new topological perspective of GNN. In the first part of the paper, the authors introduce the concept of “cellular sheaf” and “sheaf Laplacian” which allows defining a diffusive process over the graph. Interestingly, the ability of the diffusive process to correctly classify the nodes in the input labelled graph is strictly related to the class of the sheaves considered to define the process itself.

The continuous sheaf diffusion can be discretised using the Euler method to solve PDE, obtaining the sheaf convolution. The properties of the sheaf convolution are studied by observing how it affects the Dirichlet energy of the process.

In practice, the convolution sheaf operator is not known and should be learned from data. To this end, the authors propose three different convolution models that can learn the sheaf operator. In particular, the three alternatives are diagonal sheaf operators, orthogonal sheaf operators and general sheaf operators.

The experimental results show competitive performances against standard GNN models.


**Questions:**

1) Why are we interested in studying the Dirichlet energy of the sheaf convolutional network?
2) From my understanding, it seems that we prefer sheaf convolutional networks that do not converge fast to the equilibrium (the zero-energy point). On the contrary, the property of the sheaf diffusion process is studied in the time infinite limit (that should be the equilibrium). Why is there this difference?
3) How X(0) is computed? From my understanding, the node features are stored in a matrix N x F. Nevertheless, X(0) should be ND x F. Is an MLP used for this goal? Is the same MLP for all input features?
4) How is the parametric matrix-valued function \phi used? Is it used to build the sheaf Laplacian explicitly? Is the cost of evaluating \phi considered in the computational complexity (I think it should be O(d^2) in the diagonal case)?
5) The \phi defined in the paper allows building only symmetric sheaf operators. While this allows to speed up the computation, it reduces the separation power. Are there any other options that could be considered to find a trade-off between complexity and power?

Minors:
1) In proposition 1, maybe W should have subscript 1.
2) I suggest another capital letter (W is already used) to denote the parameter of the MLP which learns the sheaf operator.


**Limitations:**

Yes, the authors discuss the limitations of their work.

**Strengths And Weaknesses:**

The paper is interesting since it applies topological concepts in the context of graph learning. To the best of my knowledge, the application of the sheaf theory in this context is novel. I believe that the most interesting result is the connection between the sheaf structure and the classification ability of the diffusion process associated with that structure.

The paper is well written and easy to follow. The only part that I found not clear is Section 5 (see questions). Also, I believe that more words could be spent explaining the experimental setting (see questions).

Finally, it is hard to conclude that the proposed models outperform the baseline models due to the variance of the results. I do not believe that this reduces the impact of the paper, but I would change sentences such as “the models achieve state-of-the-art results” and similar ones.

---

> ### Author Response · Authors · 2022-07-31
> **Response to Reviewer QFM1**
>
> > **Why are we interested in studying the Dirichlet energy of the sheaf convolutional network?
> From my understanding, it seems that we prefer sheaf convolutional networks that do not converge fast to the equilibrium (the zero-energy point). On the contrary, the property of the sheaf diffusion process is studied in the time infinite limit (that should be the equilibrium). Why is there this difference?**
>
> The analysis in Section 5 is about the practical setting where one does NOT have the perfect sheaf structure, but rather the approximately right sheaf or, in the worst case, a sheaf that is completely unhelpful for the task to be solved. In this case, we would like the model to be able to avoid converging to the kernel of the Laplacian (something GCNs cannot do). The results in that section show this is indeed possible in certain conditions.
>
> However, if the right sheaf is found, then indeed we want to converge to the kernel and the model can easily do so (in the linear setting) by setting $W_1, W_2$ to identity.
>
> > **How X(0) is computed? From my understanding, the node features are stored in a matrix N x F. Nevertheless, X(0) should be ND x F. Is an MLP used for this goal? Is the same MLP for all input features?**
>
> We use a simple MLP to process the raw features. The MLP takes as input a matrix of shape $(N, F)$ and returns a matrix of shape $(N, D \times C)$, where $C$ is a hyperparameter denoting the number of channels we use during diffusion. Then this matrix is reshaped to $(N \times D, C)$. We will mention these details more explicitly in the final version.
>
> > **How is the parametric matrix-valued function \phi used? Is it used to build the sheaf Laplacian explicitly?**
>
> The function $\Phi$ is used to construct the restriction maps. We use these restriction maps to explicitly compute the (non-zero) lower triangular blocks of the Laplacian. Then, exploiting the symmetry of the Laplacian, we compute the upper triangular blocks by reflecting across the diagonal. This gives us the Laplacian, which we store as a torch sparse matrix, and we use sparse-dense matrix multiplication to compute $\Delta X(t)$.
>
> > **Is the cost of evaluating \phi considered in the computational complexity (I think it should be O(d^2) in the diagonal case)?**
>
> The cost of evaluating $\Phi$ is considered and the complexity is only $O(d)$ in the diagonal case. In the case the restriction maps are diagonal, $\Phi$ outputs only the diagonal elements of the matrix and we only work with the diagonal of the matrix (i.e. there is no need to store the zero entries). Therefore, matrix-vector and matrix-matrix multiplications can be reduced to element-wise vector multiplications and this is done in $O(d)$.
>
> > **The \phi defined in the paper allows building only symmetric sheaf operators. While this allows to speed up the computation, it reduces the separation power. Are there any other options that could be considered to find a trade-off between complexity and power?**
>
> This is a misunderstanding. $\Phi$ can learn non-symmetric restriction maps. As specified in line 271, we use $\sigma(W[x_v || x_u])$ to compute the restriction map $\mathcal{F}\_{v \trianglelefteq (v, u)}$ . It is easy to see that in general $\mathcal{F}\_{v \trianglelefteq (v, u)} = \sigma(W[x_v || x_u]) \neq \sigma(W[x_u || x_v]) = \mathcal{F}\_{u \trianglelefteq (v, u)}$. Equality is obtained only when the matrix $W$ has some special structure.
>
> More generally, full expressive power is obtained when we use an MLP with universal approximation capabilities as proven in Proposition 19. The reason we use a simpler functional form is to avoid overfitting and it is not related to the computational complexity, which is not a problem in this case.
>
> > **Finally, it is hard to conclude that the proposed models outperform the baseline models due to the variance of the results. [...] I would change sentences such as “the models achieve state-of-the-art results” and similar ones.**
>
> Indeed, the high standard deviation is a common problem for these datasets that the community has been using to evaluate heterophily and over-smoothing. This makes the comparison against the best-performing baseline (GGCN) more difficult and we will adjust our claims as suggested.
>
> > **Minors. In proposition 1, maybe W should have subscript 1. I suggest another capital letter (W is already used) to denote the parameter of the MLP which learns the sheaf operator.**
>
> Thank you for these suggestions! We will address them.

---

> > ### Comment · Reviewer_QFM1 · 2022-08-08
> > **Response to Rebuttal**
> >
> > I thank the authors for their response, which addresses most of my concerns.
> >
> > Nevertheless, I am still not sure about the computational complexity. Let us consider the diagonal case. From my understanding, the first step is building the sheaf-laplacian by:
> > 1) computing the restriction maps. The restriction map $\mathcal{F}_{v \trianglelefteq (v, u)}$ is obtained by computing $\sigma(W[x_v || x_u])$. The weight matrix $W$ has size $2d \times d$ and has no particular structure. Thus, the computation $W[x_v || x_u]$ requires $O(d^2)$ operation. There are $O(m)$ restriction maps to compute, giving the total complexity of $O(md^2)$.
> > 2) computing the sheaf-laplacian. Since all restriction maps are diagonals, their multiplication has complexity $O(d)$. Since we need to multiply $O(n)$ elements for the diagonal blocks of the sheaf-laplacian and $O(m)$ for the non-diagonal ones, the overall complexity is $O(nd+md)$.
> >
> > Thus, the overall complexity for building the sheaf-laplacian is $O(nd + md^2)$.
> >
> > Now, let us consider the complexity of the message passing of the model in Eq. 4.
> > 1) Each node $u$ has a feature matrix $X_u$ of size $d \times c$, where $c$ is the chosen number of channels.
> > 2) $X_u$ is then multiplied by $W_1$ on the left and $W_2$ on the right, where $W_1$ has size $d \times d$ and $W_2$ has size $c \times c$ (assuming that we do not change the number of channels through the layers). The multiplication on the left requires $O(d^2c)$ operation and returns a new matrix of size $d\times c$, which is then multiplied by $W_2$ requiring $O(dc^2)$ operations. Thus, we need $O(d^2c + dc^2)$ operations to compute the new feature matrix of a single node and $O(n(d^2c + dc^2))$ for the whole graphs.
> > 3) The sheaf-laplacian is used to propagate the feature nodes. Since the transportation maps are diagonals of size $d \times d$, and the feature nodes are matrices of size $d\times c$, the complexity of their product is $O(dc)$. Since we need to propagate over all the edges, we have a total complexity of $O(mdc)$.
> >
> > Thus, the overall complexity for the message passing is $O(n(d^2c + dc^2) + mdc)$.
> >
> > Summing up the two total complexity (and ignoring the number of channels), we obtain a complexity for the diagonal case that is $O(nd + md^2 + n(d^2 + d) + md) = O(d^2(m+n))$.
> >
> > I hope that this comment will help to further improve the manuscript, which is likely to be accepted given the other scores.

---

> > > ### Author Response · Authors · 2022-08-09
> > > **Response about complexity**
> > >
> > > Thank you for these clarifications! We misunderstood the point the reviewer was making originally. The analysis of the reviewer is correct. The discrepancy comes from slightly different assumptions that we clarify below. To avoid any confusion for the readers, we will include the number of feature channels explicitly in the complexity and revise the diagonal case to $O(d^2)$.
> > >
> > > For learning the restriction maps, a general function $\Phi$ has a tight lower bound of $\Omega(d)$. Here, we assumed a general MLP with a constant hidden size independent of $d$, in which case the complexity is $O(d)$. It is true, however, that for the parametrisation we use in practice (i.e. $\sigma(W[x_v || x_u])$) the complexity is $O(d^2)$. Since we use the latter layer, we will change to $O(d^2)$ as suggested to avoid confusion. In practice, there is no significant difference between the two approaches because $d$ is so small.
> > >
> > > Regarding the $W_1, W_2$ multiplications, to make the comparison with regular MPNNs easier, we did not include the complexity of the linear transformation since it has the same complexity as the linear transformation in a typical MPNN (e.g. GCN). Assuming an MPNN uses $f$ feature channels, a sheaf-based model using the same representation size satisfies $d \times c = f$. Then $d^2c + dc^2 = f(d + c) \in O(f^2)$. So the complexity of the linear transformation in our model is the same as that of a typical MPNN using $f$ feature channels. Again, to avoid any confusion, we will add the cost of this operation (both for MPNNs and SNNs) and also explicitly include $f$ and $c$ in the computational complexity

---

### Author Response · Authors · 2022-07-31
**Official Response**

We would like to thank the reviewers for their extensive reviews, helpful comments and actionable suggestions for improving our manuscript. We are delighted that the reviewers generally view our work favourably and that they deem our paper “novel” (QFM1), “​​original” (2nUo) with “timely and significant” technical contributions (s77q) that “may greatly benefit our community” (dwBG). We are also grateful they have focused on concrete suggestions for improving the manuscript, particularly in the presentation of technical details, which we will be able to act upon. Furthermore, we appreciate the insightful questions that will help improve the clarity of our discussion.

We have responded to each review directly to address their specific comments. Again, we would like to express our gratitude for these constructive reviews and welcome further comments or requests.

---

### Comment · Area_Chair_V3ZJ · 2022-08-03
**Discussion period**

Thanks to all reviewers and authors for their work on this submission.

As the discussion period starts, I want to make sure that reviewers have read the author's response.

This can be done either by communicating with authors, or in private conversation within the reviewing team.

---

### Meta-Review · Area_Chair_V3ZJ · 2022-08-20

**Recommendation:** Accept
**Confidence:** Certain

**Metareview:**

All reviewers agree that this paper deserves to be published in NeurIPS 2022, with some minor concerns that are (mostly) adressed in the rebuttal. Please incorporate the remaining reviewers' feedback for the camera-ready version.

**Award:**

No

---

### Decision · Program_Chairs · 2022-09-14

Accept